# IN-CONTEXT KV-CACHE EVICTION FOR LLMS VIA ATTENTION-GATE

## ABSTRACT

The KV-Cache technique has become the standard for the inference of large language models (LLMs). It caches states of self-attention to avoid recomputation. Yet, it is widely criticized that KV-Cache can become a bottleneck of the LLM inference system, especially when confronted with ultra-large models and long-context queries. A natural remedy is to discard the KV-Cache for less important tokens, with StreamingLLM (Xiao et al., 2024) as an example, but the used static eviction strategies cannot flexibly adapt to varying contexts. Remedies like H2O (Zhang et al., 2024) leverage accumulative attention scores to perform dynamic eviction but suffer from the attention bias issue in capturing contextual information. This paper bridges this gap by devising a parameterized KV-Cache eviction mechanism, dubbed as *Attention-Gate*, which accepts the whole context as input and yields eviction flags for each token to realize *in-context* eviction. The subsequent self-attention module proceeds according to the flags and only the KV states for the remaining tokens need to be cached. The Attention-Gates can vary among different heads and layers and be trivially plugged into pre-trained LLMs, tuned by cost-effective continual pre-training or supervised fine-tuning objectives to acquire what to discard. The computational and memory overhead introduced by Attention-Gates is minimal. Our method is validated across multiple tasks, demonstrating both efficiency and adaptability. After a highly efficient continual pre-training, it achieves higher average accuracy and evicts more tokens compared to traditional training-free methods. In supervised fine-tuning, it not only evicts many tokens but also outperforms LoRA-finetuned LLMs on some datasets, such as RTE, where it improves accuracy by 13.9% while evicting 62.8% of tokens, showing that effective eviction of redundant tokens can even enhance performance.

## 1 INTRODUCTION

Large language models (LLMs) (Dubey et al., 2024; Team et al., 2024; Chiang et al., 2023) have achieved remarkable success across a wide range of natural language processing tasks. A key technique that has enabled efficient LLM inference is KV-Cache, which stores transient attention keys and values to avoid recomputation. However, as the size of LLMs continues to increase and the demand for handling long-context queries grows, the KV-Cache has emerged as a significant bottleneck. Storing attention states for numerous tokens can lead to considerable memory overhead and data transfer among the memory hierarchy results in substantially increased inference time.

Studies have shown that sparsity is a natural phenomenon in attention mechanisms, with many tokens being redundant for inference (Zhang et al., 2024). This suggests that retaining all tokens in the KV-Cache is unnecessary. Existing works have explored this insight to compress KV-Cache using static strategies or hinging on accumulative attention scores. StreamingLLM (Xiao et al., 2024) is a representative of the former by retaining a fixed window of beginning and recent tokens in the KV-Cache but it struggles to flexibly adapt to specific contexts. E.g., in sentiment analysis, retaining the token "cute" in "a cute cat" is crucial, while in object recognition, the token "cat" would be more important. H2O (Zhang et al., 2024), on the other hand, employs a token-adaptive approach, using local accumulative attention scores to determine which tokens to evict. However, it is criticized that in practice, H2O suffers from the attention bias issue (Oren et al., 2024), with a tendency to over-prioritize either the initial or recent tokens.

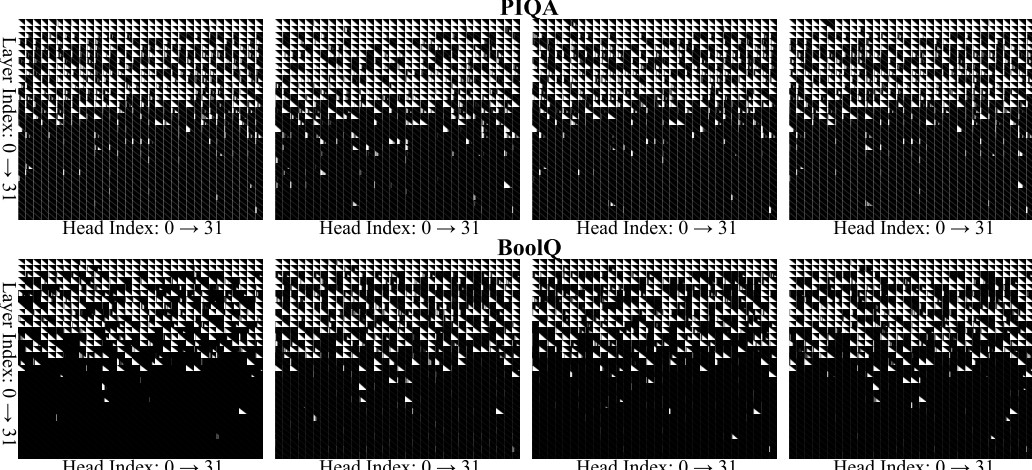

Figure 1: KV-Cache eviction patterns across different layers and attention-heads, visualized for 4 samples from the PIQA dataset (**top row**) and 4 samples from the BoolQ dataset (**bottom row**), using AG fine-tuned Llama2-7B models. Black areas represent tokens that are neither computed nor stored in the KV-Cache. The variability of eviction patterns across tasks, prompts, layers, and attention-heads demonstrates the dynamic nature of our method. A common trend observed is that deeper layers tend to mask more KV-Cache states, with some in deeper layers being entirely masked.

To overcome these challenges, we introduce a parameterized KV-Cache eviction mechanism named the Attention-Gate (AG), to perform reliable *in-context* eviction. AG is positioned before a self-attention layer within the model. It takes a sequence of token features as input and generates eviction flags for the tokens, indicating whether a token should be excluded from subsequent self-attention computations. Tokens that are evicted do not require their KV states to be cached. AGs can be seamlessly integrated into pre-trained LLMs and tuned by minimizing the language modeling loss. Ideally, AGs can automatically learn to discern the most relevant tokens for the current context without manual intervention. In practice, we can implement the AG as a self-attention layer with much fewer heads than the original model (e.g., 4 v.s. 32). This way, the parallel computational capabilities of the hardware can be harnessed to minimize the extra overhead introduced by AGs.

AG is empirically shown to enjoy high training efficiency, e.g., only four NVIDIA 4090 GPUs and a dataset of 5,000 samples are required for continual pre-training when applying AGs to LLaMA2-7B (Touvron et al., 2023). This alleviates concerns about the computational overhead related to trainable eviction strategies (Zhang et al., 2024; Chen et al., 2024) and amplifies the performance superiority of our approach over existing training-free approaches. As illustrated in Figure 1, AG generates different eviction strategies across different layers and attention-heads for different tokens, demonstrating its adaptability to the diverse requirements of each component in the model.

To validate the effectiveness of our method, we conduct extensive experiments across multiple benchmarks. After efficient continual pre-training (CPT), our approach outperforms traditional training-free eviction strategies, such as StreamingLLM and H2O, in both accuracy and token eviction rates. In supervised fine-tuning (SFT), our method not only evicts a significant number of redundant tokens but also maintains or surpasses the performance of LoRA-finetuned LLMs. For example, on the RTE dataset, our approach improves accuracy by 13.9% while evicting 62.8% of tokens, demonstrating that selective token eviction can enhance performance. In summary, the Attention-Gate mechanism provides a scalable and efficient solution for KV-Cache management, addressing the limitations of traditional training-free methods.

## 2 RELATED WORK

As large language models (LLMs) scale in both size and input sequence length, optimizing their efficiency has become increasingly important, particularly in addressing space and time complexity.

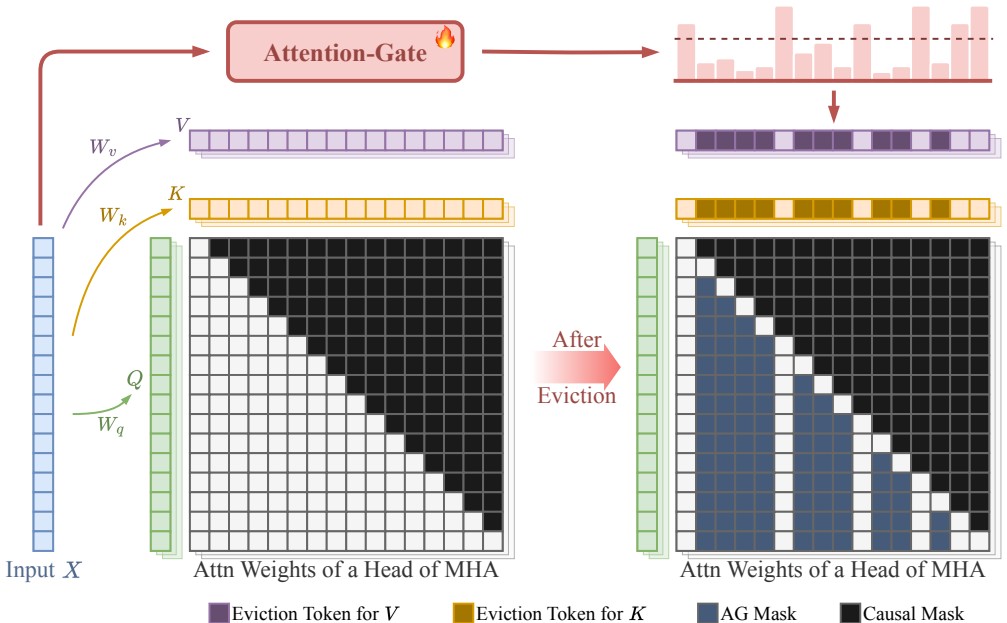

Figure 2: An overview of Attention-Gate (AG) for KV-Cache eviction. AG is a lightweight learnable module placed before each MHA layer. Given the input hidden states, it determines for each head whether to retain or discard the key and value tokens in the KV-Cache. In the attention weights, this corresponds to masking out columns for the evicted keys, while keeping the diagonal intact to ensure the query interacts with its own key.

A significant bottleneck lies in the attention mechanism, which demands considerable computational and memory resources, especially for long sequences.

**Computation and memory challenges.** The computational cost of attention is driven by its quadratic complexity, while the memory burden stems from the need to store the KV-Cache during inference to avoid recomputation. Although several works have focused on reducing the computational load of attention mechanisms, many fail to address the memory constraints that remain a critical issue. Techniques like Reformer (Kitaev et al., 2020) and FlashAttention (Dao et al., 2022; Dao, 2023) mitigate memory usage for long sequences but still require substantial cache storage. On the other hand, approaches such as MQA (Shazeer, 2019) and GQA (Ainslie et al., 2023) prioritize memory compression without tackling computational complexity. These solutions tend to focus on either memory or computational efficiency, rarely addressing both simultaneously.

**KV-Cache eviction strategies.** To address both memory and computational challenges, KV-Cache eviction has emerged as an effective strategy. Existing approaches can be categorized into static and adaptive methods based on accumulative attention scores.

Static strategies, such as those used in Sparse Transformers (Child et al., 2019), employ fixed pruning patterns, such as Strided and Fixed Attention. While effective in some cases, these approaches are not adaptive to specific contexts, often sacrificing accuracy. StreamingLLM (Xiao et al., 2024) tackles the *Attention Sink* phenomenon, where attention scores concentrate on initial tokens, by retaining these tokens along with a fixed window of recent tokens. While this improves performance, static approaches generally lack the flexibility needed to adapt to different tokens, attention-heads, or layers.

Strategies using accumulative attention scores offer more flexibility by dynamically identifying important tokens. For instance, SpAtten (Wang et al., 2021) employs Accumulative Attention Scores (A2S), which sum the softmax outputs for each token to measure its importance. This approach allows selective token pruning in subsequent layers, effectively reducing computational complexity without the need for retraining. H2O (Zhang et al., 2024) extends this concept to decoder-based models, using local A2S statistics for adaptive eviction in autoregressive generation. However, H2O suffers from the attention bias issue (Oren et al., 2024), particularly in long-context inputs. Several

follow-up works have aimed to address this limitation. NACL (Chen et al., 2024) introduces random eviction to mitigate attention bias, while A2SF (Jo & Shin, 2024) incorporates a Forgetting Factor. However, none of these approaches fully resolve the underlying problem.

**More Adaptive strategies.** Although strategies based on accumulative attention scores provide more flexibility than static methods, they still have notable limitations. For instance, H2O (Zhang et al., 2024) applies the same token eviction ratio across all attention heads, restricting the adaptability of the method. FastGen (Ge et al., 2023), on the other hand, introduces a different approach by hybridizing KV-Cache compression policies and applying adaptive strategies to each attention head. However, it focuses on the decoding stage and neglects the importance of the prefilling stage. Learnable eviction strategies, on the other hand, offer greater flexibility by enabling different layers and attention heads to adopt heterogeneous eviction policies. However, such strategies have been relatively underexplored, likely due to concerns about the computational overhead they may introduce (Zhang et al., 2024; Chen et al., 2024). Nonetheless, task-specific training is essential for optimizing performance across different contexts. For example, a recent approach (Anagnostidis et al., 2024) introduces a learnable mechanism for dropping uninformative tokens, but it faces difficulties in batched generation and does not account for continual pre-training or decoding-only LLMs. Despite these challenges, learnable strategies have strong potential to improve performance across a variety of tasks by allowing models to adapt their eviction strategies to meet task-specific requirements.

## 3 METHOD

This section first briefly reveals the basics of multi-head attention and KV-Cache and then describes the proposed *Attention-Gate* (AG) mechanism for in-context KV-Cache eviction for LLM inference acceleration. An illustrative overview of AG is presented in Figure 2. For additional supplementary descriptions of AG, see Appendix A.

### 3.1 PRELIMINARY

**Multi-Head Attention** (MHA) (Vaswani et al., 2017) is a core component of the Transformer architecture, widely used in most LLMs. MHA enables the model to capture dependencies across different tokens in a sequence by computing self-attention across multiple attention-heads. Each head attends to different parts of the input sequence independently, allowing the model to capture various aspects of the relationships between tokens. The outputs of these attention heads are then concatenated and projected through a final output matrix.

**KV-Cache** is employed to store the key and value representations of tokens from previous time steps during inference. This prevents redundant recomputation of these representations for every new token generated, significantly speeding up auto-regressive generation. The inference process in auto-regressive transformers, such as Llama (Touvron et al., 2023), can be divided into two stages: *prefilling* and *decoding*. **1.** In the prefilling stage, the model processes the entire input sequence, generating key-value pairs for each attention-head in each layer. These key-value pairs are stored in the KV-Cache for future reuse. **2.** In the decoding stage, when generating the next token, the model only computes the key and value for that new token and appends them to the cached key-value pairs. This reduces the computational load, as the previously cached key-value pairs do not need to be recomputed. The process continues token by token until the sequence generation is complete.

KV-Cache plays a critical role in improving the efficiency of LLM inference, especially in scenarios where long sequences are processed. However, the size of the KV-Cache grows with the input sequence length, leading to substantial memory overhead. Efficiently managing this cache while maintaining model performance has become a key challenge in scaling LLMs to longer contexts.

### 3.2 LIMITATIONS OF TRADITIONAL EVICTION STRATEGIES

**Lack of flexibility.** Flexibility in KV-Cache eviction strategies spans across various dimensions, including token-specific, attention-head-specific, layer-specific, task-specific, and model-specific adaptability. Static eviction policies offer no such flexibility, requiring manual intervention to adjust them. While methods based on accumulative attention scores (e.g., H2O (Zhang et al., 2024)) have improved upon static strategies by introducing token-level and head-level adaptability, they are still

limited. Specifically, in H2O, the eviction ratio remains uniform across all attention heads, restricting head-level flexibility. Flexibility is crucial in eviction strategies because models process varying types of data across different contexts, requiring the ability to selectively retain important tokens while discarding others. Without this adaptability, the model may inefficiently store redundant information, increasing memory usage and slowing down inference times.

**Absence of global statistics.** Eviction strategies that rely on accumulative attention scores, such as H2O, base their calculations on local statistics, which often results in attention bias issues (Oren et al., 2024). This local approach may misjudge a token's importance, especially when the context spans long sequences. While several methods (Chen et al., 2024; Jo & Shin, 2024) have proposed solutions to mitigate this bias, they are still fundamentally rooted in local statistics, addressing the issue only partially. A more effective solution would incorporate global statistics to ensure that the eviction process is based on a more comprehensive understanding of the context, thereby reducing biases and improving the accuracy of token retention.

**Inefficiency.** Efficiency is another challenge for traditional methods such as H2O and FastGen (Ge et al., 2023). These methods operates in a token-by-token manner for each decoding step, sequentially deciding which key-value pairs to evict from the cache. Additionally, H2O determines which tokens to evict after computing the attention scores, meaning some computation is wasted on tokens that are later discarded. This sequential approach hinders efficiency, especially when handling large batches of data or when processing long sequences.

Our Attention-Gate mechanism is designed to address the aforementioned limitations.

### 3.3 ATTENTION-GATE

The *Attention-Gate* (AG) is a lightweight, trainable module designed to determine which tokens in the KV-Cache of each attention-head should be retained or discardedAG is positioned before the Multi-Head Attention (MHA) layer and operates by generating *eviction flags* for each token, guiding both the computation of attention scores and the management of the KV-Cache.

**Input and output.** AG takes the hidden states as input and generates binary flags as output, one for each token in every attention head. These flags indicate whether the corresponding token's key-value pairs should be stored or discarded in the KV-Cache. Specifically, the input is a hidden state matrix $X \in \mathbb{R}^{n \times d}$, where $n$ represents the sequence length, and $d$ denotes the dimensionality of the hidden states. The output consists of independent binary decisions for the retention or eviction of each token's key-value pairs across all attention heads in the MHA module.

**Local or global?** The structure of AG offers two choices: using either local or global information to guide the eviction policy. **1.** Local information: The simplest approach leverages a *linear layer* where each token only considers its own hidden state without taking other tokens in the sequence into account. This method is computationally efficient and straightforward, making it appealing for its simplicity. **2.** Global information: On the other hand, a more comprehensive approach adopts an attention-like structure, where tokens can aggregate information from across the entire sequence. This enables the eviction policy to reflect the global in-context information, making decisions that are informed by the broader context. Naturally, this method is computationally heavier than the local approach, but it offers a more accurate reflection of the sequence's token importance.

In our design, we opted for the attention-like structure because *global context is essential for making effective eviction decisions*. As demonstrated in Section 4.4, using an attention-like structure significantly outperforms the linear layer, as the latter struggles to properly evict tokens. This outcome is expected, as determining whether a token is redundant or crucial for inference often requires a global understanding of the sequence. A local approach, like a linear layer focusing on individual tokens, lacks the broader perspective necessary to make accurate decisions about which tokens are important. This finding also underscores the limitations of methods like H2O, which rely on local statistics rather than global ones, as mentioned in Section 3.2.

**Softmax or Sigmoid?** After gathering information through the attention-like structure, the next step is to compute the eviction probability for each token. Common activation functions such as *Softmax* and *Sigmoid* can be used to generate these probabilities.

In our case, we choose the Sigmoid function for a key reason: while Softmax normalizes probabilities across the entire sequence, introducing competition between tokens, Sigmoid treats each token independently. This independence is crucial because we do not want a token's likelihood of being evicted to depend on the eviction probabilities of other tokens in the sequence.

**Eviction flags.** After computing the eviction probabilities for each token, we apply a threshold $\tau$. If the probability of retaining a token exceeds $\tau$, that token is retained in the KV-Cache; otherwise, it is discarded. In the attention matrix, the columns corresponding to evicted tokens are masked out. However, the diagonal elements, where a token attends to itself, are always preserved. This guarantees that each token continues to interact with its own key.

**Prefilling or decoding?** AG is primarily applied during the prefilling stage, where the full sequence is available. By making eviction decisions before the MHA layers, AG effectively manages the KV-Cache during this phase. In contrast, AG is not used during the decoding stage to avoid adding complexity to the inference process. Using AG in decoding could slow down inference and increase the cache memory footprint, as additional keys and values from the attention-like structure would need to be stored.

In summary, AG dynamically determines which tokens should be retained, optimizing both KV-Cache usage and attention score computation by utilizing an attention-like structure, Sigmoid activation function, and a threshold-based decision for token eviction.

### 3.4 TRAINING IMPLEMENTATION

To train the Attention-Gate (AG) effectively, this section outlines the key components of the training process. For more comprehensive details, please refer to Section 4 and Appendix B.

**Eviction Loss.** To encourage the eviction of unnecessary tokens, we introduce a dedicated loss function called the *Eviction Loss*. This loss encourages the model to discard as many tokens as possible, which is defined as:

$$\ell_{\text{evict}} = \alpha \cdot \left| \overline{\text{AG}} - \beta \right|, \tag{1}$$

where $\overline{\text{AG}}$ represents the average output of all AG modules. In this formula, $\alpha$ adjusts the intensity of KV-Cache eviction, while $\beta \in [0, \tau]$ ensures that eviction does not become overly aggressive. This loss function works alongside the auto-regressive loss to balance token eviction with maintaining model performance. The loss allows for adaptive eviction across layers and attention-heads. As shown in Figure 1, after training, the model learns to increase token eviction in deeper layers, which is not achievable with training-free methods like H2O (Zhang et al., 2024) .

**Initialization.** We initialize the AG parameters using Xavier initialization (Glorot & Bengio, 2010) to provide a stable starting point for learning. Additionally, a small constant $\gamma \geq 0$ is added inside Sigmoid, ensuring that the initial retention probabilities are close to 1. This encourages the model to retain most tokens early in training, allowing it to learn which tokens are important more gradually.

**Handling non-differentiability.** Directly applying the threshold-based gating mechanism from Section 3.3 would lead to non-differentiable gradients during training due to the hard thresholding's discrete nature. To resolve this, we employ the *Straight-Through Estimator* (STE) (Yin et al., 2019), which allows gradients to flow through discrete decisions by approximating them during the backward pass. Specifically, during backpropagation, instead of using the hard 0 or 1 values obtained from comparing against the threshold, we utilize the smooth output of the Sigmoid function. This approach ensures smooth gradients and enables effective training of the AG while preserving its binary behavior during the forward pass.

## 4 EXPERIMENTS

This section consists of three main parts. First, we evaluate the performance of AG in two scenarios: continual pre-training (CPT) and supervised fine-tuning (SFT). For CPT, we compare AG with classic methods on performance and eviction rates (Section 4.1). For SFT, we benchmark AG against vanilla LoRA fine-tuning to highlight its task-specific adaptability (Section 4.2). Second, we provide visualization of selected examples to demonstrate the core characteristics of AG (Section 4.3). Finally,

Table 1: Performance comparison of Llama2-7B and various KV-Cache eviction strategies. Our approach trains only the AG module during continual pre-training, keeping other components frozen. The table reports accuracy (Acc.) for Llama2-7B and all eviction methods, with Llama2-7B serving as the upper bound for accuracy. Metric %Eviction refers to the mean KV-Cache eviction ratio, representing the percentage of tokens evicted from the KV-Cache. The eviction ratio is fixed at 50% for the baseline methods, including a local strategy (retaining recent tokens), StreamingLLM, and H2O. In contrast, our method achieves higher average accuracy alongside a higher average %Eviction.

|  | Metric | PIQA | ARC-C | ARC-E | RTE | COPA | BoolQ | HellaSwag | Avg. |
|---|---|---|---|---|---|---|---|---|---|
| Llama2-7B | Acc. | 76.33 | 37.29 | 51.32 | 51.99 | 62.00 | 69.94 | 68.16 | 59.58 |
| Local | Acc. | 69.97 | 31.86 | 48.68 | 51.99 | 60.00 | 57.86 | 37.08 | 51.06 |
| StreamingLLM | Acc. | 72.69 | 33.22 | 51.15 | 50.18 | **63.00** | 62.05 | 40.85 | 53.31 |
| H2O | Acc. | 75.90 | 33.22 | **52.03** | **52.71** | 47.00 | 67.37 | 66.32 | 56.36 |
| Ours | Acc. | **76.17** | **33.90** | 49.03 | 52.35 | **63.00** | 67.52 | **66.33** | **58.33** |
|  | %Eviction | **54.29** | **51.03** | **51.05** | 46.70 | 40.02 | **57.75** | **52.16** | **51.87** |

Table 2: Performance of Llama2-7B with LoRA fine-tuning and our method on six downstream tasks. In addition to the LoRA fine-tuning targets, our method makes the AG modules learnable. Two settings for $\alpha$ (0.5 and 1) are tested. Our method maintains comparable or better accuracy while achieving a higher eviction ratio, demonstrating its task-specific adaptability in managing token eviction without significant accuracy loss.

|  | Metric | PIQA | ARC-C | RTE | COPA | BoolQ | OBQA | Avg. |
|---|---|---|---|---|---|---|---|---|
| Fine-tuned Llama2-7B | Acc. | 82.92 | 60.34 | 64.98 | 92.00 | 88.10 | 78.80 | 77.86 |
| Ours ($\alpha = 1$) | Acc. | 82.15 | 59.66 | 64.26 | 93.00 | 86.82 | 78.80 | 77.45 |
|  | %Eviction | 66.16 | 48.31 | 65.47 | 45.40 | 67.46 | 67.17 | 60.00 |
| Ours ($\alpha = 0.5$) | Acc. | 81.50 | 57.63 | 74.01 | 95.00 | 87.00 | 79.20 | 79.06 |
|  | %Eviction | 64.96 | 36.45 | 62.80 | 34.77 | 67.31 | 66.65 | 55.49 |

we conduct ablation studies to provide further insights into the effectiveness of AG (Section 4.4). Additional results and analysis for CPT are provided in Appendix B.1.

## 4.1 CONTINUAL PRE-TRAINING

### 4.1.1 SETUP

**Models and datasets.** We use Llama2-7B (Touvron et al., 2023) as our primary base model due to its strong performance and popularity within the AI community. Additionally, we validate the feasibility of our approach on Mistral-7B (Jiang et al., 2023), with results provided in Table 5. For continual pre-training, we select a subset of the Redpajama dataset (Computer, 2023) containing approximately 5,000 samples [1]. To assess the effectiveness of our method, we evaluate it on seven widely recognized benchmarks: PIQA (Bisk et al., 2020), ARC-C (Clark et al., 2018), ARC-E (Clark et al., 2018), RTE (Bar-Haim et al., 2006), COPA (Roemmele et al., 2011), BoolQ (Clark et al., 2019), and HellaSwag (Zellers et al., 2019). All evaluations are conducted in a zero-shot setting, with accuracy assessed using OpenCompass (Contributors, 2023).

**Training details.** During continual pre-training (CPT), only the AG modules are learnable, while the rest of the model's parameters are frozen. For the Llama2-7B model, the hyperparameters from Section 3.4 are used: $\tau = 0.5$, $\gamma = 2$, $\alpha = 5$, and $\beta = 0.4$. The model was trained for one epoch.

**Metrics and Baselines.** The metrics considered include accuracy and the mean eviction ratio for all KV-Cache. The accuracy of vanilla Llama2-7B on these datasets is used as the upper bound, with the goal of minimizing accuracy loss while maximizing the mean eviction ratio. For KV-Cache eviction strategies, we select a local method (retaining only recent tokens) and StreamingLLM (Xiao et al., 2024) as representatives of static strategies, while H2O (Zhang et al., 2024) serves as a representative of methods based accumulative attention scores. For all these methods, eviction ratio is set to 50%.

---

[1] We sampled 4,997 samples proportionally from each subset of RedPajama.

### 4.1.2 RESULTS

The results in Table 1 demonstrate the effectiveness of our method in balancing accuracy and KV-Cache eviction. Our method consistently outperforms the baseline strategies in terms of accuracy across most tasks while achieving a higher mean eviction ratio. This indicates that our method is able to evict more tokens on average without significantly compromising accuracy.

In terms of accuracy, our method achieves competitive performance, closely matching the performance of vanilla Llama2-7B in most cases. For instance, on PIQA, COPA, and BoolQ, our method performs comparably to Llama2-7B, demonstrating minimal accuracy degradation. Compared to baselines, our method shows superior performance, particularly in tasks like BoolQ and PIQA, where both accuracy and eviction ratio surpass the baselines. This confirms the advantage of learnable mechanism for more efficient token retention strategies without sacrificing model performance.

Moreover, it is worth highlighting that our method achieves these results with minimal computational overhead, as mentioned in Section 4.1.1. The continual pre-training was conducted on only 5,000 samples and trained for just one epoch, demonstrating the lightweight nature of our approach. This efficiency can be attributed to the fact that our method does not need to learn new knowledge from scratch but rather focuses on learning effective token retention strategies, leveraging the existing capabilities of the pre-trained model.

## 4.2 SUPERVISED FINE-TUNING

### 4.2.1 SETUP

**Model and tasks.** We utilize Llama2-7B (Touvron et al., 2023) as base model. To evaluate our approach, we select six widely recognized downstream tasks: PIQA (Bisk et al., 2020), ARC-C (Clark et al., 2018), RTE (Bar-Haim et al., 2006), COPA (Roemmele et al., 2011), BoolQ (Clark et al., 2019), and OpenBookQA (Mihaylov et al., 2018). For each task, we fine-tune the model using the respective training set and evaluate its performance on the corresponding test set.

**Baselines and Implementation Details.** For the baseline, we apply LoRA (Hu et al., 2021) to fine-tune the models on each task, targeting the $(W_q, W_k, W_v, W_o)$ weights in the self-attention modules. In our method, in addition to applying LoRA to these parameters, the AG modules are also set to be learnable. As mentioned in Section 3.4, the following hyperparameters are used: $\tau = 0.3$, $\gamma = 0$, $\alpha = 1$ or $0.5$, and $\beta = 0.28$. We employ the AdamW optimizer (Loshchilov & Hutter, 2017) with a learning rate of 5e-5, and train for two epochs on each dataset.

### 4.2.2 RESULTS

As shown in Table 2, our method demonstrates a strong balance between accuracy and KV-Cache eviction across the six downstream tasks.

With $\alpha = 1$, our method maintains competitive accuracy compared to the fine-tuned Llama2-7B baseline, with minimal drops (e.g., 82.15% vs. 82.92% on PIQA), while achieving a high mean eviction ratio of 60.00%. With $\alpha = 0.5$, the eviction rate is reduced to 55.49%. But in some tasks like RTE and COPA, our method even surpasses the baseline accuracy. The average accuracy also exceeds the baseline (79.06% vs. 77.86%), suggesting that effectively evicting redundant tokens allows the model to focus on relevant information and improves performance.

Additionally, under the same hyperparameter settings, the performance varies across tasks. For instance, ARC-C is more challenging to evict compared to OpenBookQA, leading to a larger accuracy drop post-eviction. This highlights the importance of task-specific KV-Cache eviction policies.

## 4.3 VISUALIZATION

In this section, we present visualizations to highlight key characteristics of our AG mechanism. After fine-tuning on specific tasks, we visualize the model's MHA and AG weights for selected samples, as shown in Figure 3. Additional visualizations can be found in Appendix B.3.

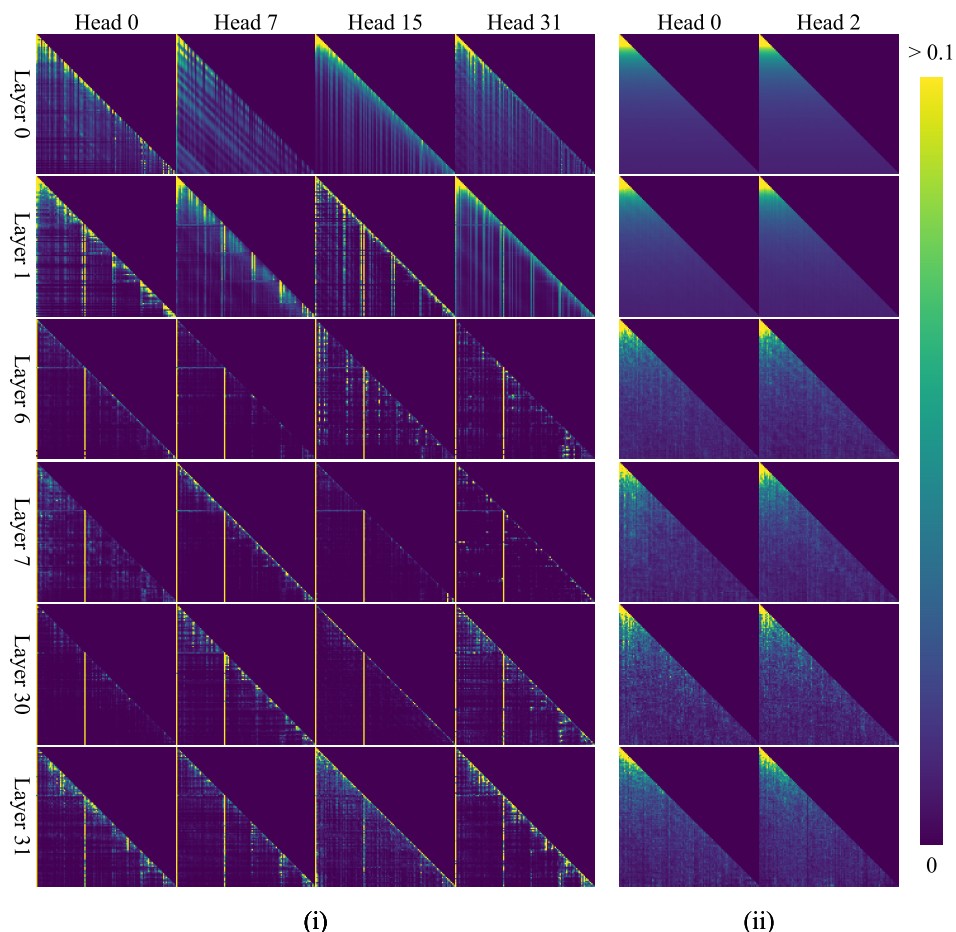

Figure 3: This visualization highlights attention patterns in Llama2-7B after fine-tuning on the BoolQ dataset, using a selected sample to showcase multiple heads within both MHA and AG of different layers. In part (i), we visualize attention scores from several MHA heads across different layers before eviction. **1.** MHA heads exhibit diverse attention patterns, especially in the first two layers, where attention is highly heterogeneous across heads. **2.** While the first two layers show dense attention patterns, the subsequent layers become progressively sparser. **3.** Bright-yellow vertical lines consistently appear at the same position across nearly all heads, especially beyond the first two layers, indicating critical tokens for inference. This aligns with the *Heavy Hitters* in H2O (Zhang et al., 2024), where a small portion of tokens significantly contributes to the attention scores. Our method's effectiveness is demonstrated by the persistence of these bright lines in deeper layers, showing that crucial tokens are retained rather than evicted, ensuring their importance is preserved across the network. In part (ii), we visualize the attention scores from the attention-like structure within the AG mechanism. As the model layers deepen, the attention pattern shifts from high-resolution to lower-resolution, indicating that AG increasingly focuses on distilling in-context information. Thus, deeper AG layers no longer need such high resolution to capture global information, as the model has already refined and summarized it. This suggests potential optimizations, such as reducing the number of heads or dimensionality in deeper AG layers, to further improve efficiency.

## 4.4 ABLATION

In this ablation study, we explore the effects of various configurations on the AG mechanism, focusing on the number of AG heads, AG head dimensions, and eviction strategies. For details of each setting and the corresponding performance, refer to Table 3. Key findings include the impact of reducing the number of heads, as shown in (2-1) and (2-2), and the reduction of head dimensions in (3-1) and (3-2), both of which result in lower accuracy and eviction ratios. In (4-1) and (4-2), we examine

Table 3: Ablation study on various settings of AG, reporting accuracy (Acc.) and KV-Cache eviction ratio (%Eviction) under different configurations, with $\alpha = 1$ in all settings. The results for (1) correspond to the setup described in Section 4.2, where the number of AG heads is 4, and the head dimension is 128. In (2-1) and (2-2), we explore the impact of the number of AG heads, with (2-1) using 2 heads and (2-2) using 1 head. The comparison between (1), (2-1), and (2-2) shows that reducing the number of heads leads to a drop in both accuracy and eviction ratio, indicating that the capacity of AG is closely tied to the number of heads. For (3-1) and (3-2), we assess the effect of reducing head dimensions for AG heads, where (3-1) has half the dimension size of (1) and (3-2) has 1/4. Comparing (1), (3-1), and (3-2) reveals that smaller dimensions reduce the eviction capability and accuracy, highlighting the importance of maintaining sufficient head dimensionality. Settings (4-1) and (4-2) examine using the hidden states and the AG module from the previous layer to inform the current layer's eviction strategy. In (4-1), the first layer does not evict, and from the second layer onward, the eviction is determined by the previous layer. In (4-2), the first and second layers do not evict, following the observation from Section 4.3 that the first two layers are denser than subsequent layers. Both (4-1) and (4-2) show a slight decline in accuracy and eviction ratio compared to (1), but this approach introduces parallelism, offering a potential avenue for future optimizations. The setting (5) replaces the attention-like structure in AG with a simple linear layer to determine the eviction strategy. The comparison with (1) shows that linear layers almost cannot evict tokens effectively, reinforcing the necessity of leveraging global in-context information for successful eviction, as discussed in Section 3.3.

| | Metric | PIQA | ARC-C | RTE | COPA | BoolQ | OpenBookQA | Avg. |
|---|---|---|---|---|---|---|---|---|
| (1) | Acc. | 82.15 | 59.66 | 64.26 | 93.00 | 86.82 | 78.80 | 77.45 |
| | %Eviction | 66.16 | 48.31 | 65.47 | 45.40 | 67.46 | 67.17 | 60.00 |
| (2-1) | Acc. | 81.88 | 57.63 | 65.70 | 91.00 | 87.52 | 77.40 | 76.86 |
| | %Eviction | 63.92 | 36.38 | 62.73 | 24.38 | 65.22 | 63.57 | 52.70 |
| (2-2) | Acc. | 82.15 | 53.90 | 62.45 | 89.00 | 87.31 | 77.40 | 75.37 |
| | %Eviction | 58.97 | 31.47 | 59.77 | 20.32 | 63.02 | 59.17 | 48.79 |
| (3-1) | Acc. | 81.45 | 53.36 | 58.84 | 88.00 | 86.73 | 78.40 | 74.46 |
| | %Eviction | 61.75 | 33.55 | 61.34 | 19.24 | 64.59 | 59.59 | 50.01 |
| (3-2) | Acc. | 83.03 | 53.90 | 59.93 | 89.00 | 87.16 | 76.40 | 74.90 |
| | %Eviction | 58.68 | 24.23 | 32.23 | 12.28 | 59.40 | 55.54 | 40.39 |
| (4-1) | Acc. | 81.66 | 55.25 | 66.06 | 88.00 | 86.85 | 78.00 | 75.97 |
| | %Eviction | 49.52 | 36.92 | 46.85 | 28.74 | 56.02 | 60.32 | 46.40 |
| (4-2) | Acc. | 82.75 | 55.93 | 79.06 | 82.00 | 86.33 | 78.40 | 77.41 |
| | %Eviction | 53.31 | 44.38 | 51.20 | 47.95 | 61.98 | 61.73 | 53.43 |
| (5) | Acc. | 82.54 | 54.58 | 57.40 | 81.00 | 87.71 | 74.80 | 73.01 |
| | %Eviction | 1.06 | 0.46 | 0.81 | 0.26 | 1.38 | 1.16 | 0.86 |

eviction strategies where the current layer's eviction is based on the previous layer's hidden states and AG module, which introduces parallelism but slightly reduces performance. Finally, replacing the AG with a linear layer in (5) demonstrates the necessity of using an attention-like structure for effective token eviction. For more ablation, please refer to Table 6.

## 5 CONCLUSION

In conclusion, the proposed Attention-Gate mechanism offers a flexible and adaptive solution to KV-Cache eviction in large language models. By dynamically identifying and discarding less important tokens, Attention-Gate addresses the limitations of static and attention-score-based strategies, providing efficient context-aware eviction. This mechanism integrates seamlessly with pre-trained models and can be easily tuned, making it a practical and effective method for enhancing both performance and memory efficiency in various tasks.

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

# A SUPPLEMENTARY DESCRIPTION OF AG

**Multi-Head Attention** (MHA) (Vaswani et al., 2017) is a core component of the Transformer architecture, as used by most LLMs. It allows the model to capture dependencies between different tokens in a sequence. Specifically, given an input sequence $X \in \mathbb{R}^{n \times d}$, the output of MHA is computed as:

$$\text{MHA}(X) = [\text{H}_1(X), \text{H}_2(X), \ldots, \text{H}_h(X)] W^O , \tag{2}$$

where $\{\text{H}_i\}_{i=1}^{h}$ refers to the $h$ attention heads and

$$\text{H}_i(X) = \text{Attn}\left(XW_i^Q, XW_i^K, XW_i^V\right) = \text{Attn}\left(Q_i, K_i, V_i\right) , \tag{3}$$

$$= \text{Softmax}\left(\frac{Q_i K_i^\top}{\sqrt{d_k}} - \text{INF}(\mathbb{1} - M_i)\right) V_i = A_i V_i. \tag{4}$$

Here, $W_i^Q, W_i^K \in \mathbb{R}^{d \times d_k}$, $W_i^V \in \mathbb{R}^{d \times d_v}$, and $W^O \in \mathbb{R}^{hd_v \times d}$ are learned projection matrices. INF is a large constant, $\mathbb{1}$ is a matrix of ones, $M_i$ is the mask applied to head $\text{H}_i$, and $A_i$ represents the attention scores for head $\text{H}_i$.

**KV-Cache** is employed during the inference of auto-regressive transformers, which stores the key and value information from previous time steps, allowing efficient reuse and reducing recomputation. The inference process can be divided into two stages: prefilling and decoding.
In the prefilling stage, the input sequence $X^{(\leq n)} = \left[x^{(1)}, x^{(2)}, \ldots, x^{(n)}\right] \in \mathbb{R}^{n \times d}$ passes through MHA, and the corresponding key-value pairs $K_i^{(\leq n)}$ and $V_i^{(\leq n)}$ for head $\text{H}_i$ are stored in the KV-Cache. These are expressed as:

$$K_i^{(\leq n)} = \left[k_i^{(1)}, k_i^{(2)}, \cdots, k_i^{(n)}\right] , \quad V_i^{(\leq n)} = \left[v_i^{(1)}, v_i^{(2)}, \cdots, v_i^{(n)}\right] ,$$

where $k_i^{(t)} = x^{(t)} W_i^K$ and $v_i^{(t)} = x^{(t)} W_i^V$. After prefilling, the next token $x^{(n+1)}$ is generated.
In the decoding stage, $x^{(n+1)}$ is input to generate $x^{(n+2)}$ for the first step. During this process, only $k_i^{(n+1)}, v_i^{(n+1)}$ need to be computed for each head $\text{H}_i$. These are then concatenated with the cached $K_i^{(\leq n)}$ and $V_i^{(\leq n)}$ to form $K_i^{(\leq n+1)}$ and $V_i^{(\leq n+1)}$, which are used to complete the current step's MHA computation and update the KV-Cache. The process repeats token by token until the sequence generation is complete.

**Attention-Gate** (AG) is introduced to capture global in-context information and determine which tokens to retain or discard. AG consists of (i) an attention-like structure and (ii) a gating mechanism. (i) MHA$'$, a modified version of the standard MHA in Equation (2), is used to facilitate global information exchange across all tokens in the sequence. To distinguish it from the vanilla MHA, all symbols in MHA$'$ are marked with a prime ($'$). Notably, MHA$'$ has fewer heads than MHA, i.e., $h' < h$, and its projection matrix $W^{O'} \in \mathbb{R}^{h' d_k' \times h}$ differs slightly from $W^O$. (ii) The gating mechanism, denoted as Gate G, is then introduced to create different eviction policies for the attention heads in the MHA layer.

Given an input sequence $X \in \mathbb{R}^{n \times d}$, the output of AG is computed as:

$$\text{AG}(X) \in \mathbb{R}^{n \times h} = \text{G}\left(\text{MHA}'(X), \tau\right) , \tag{5}$$

For a token $x$ in sequence $X$, the output of gate G is defined as:

$$\text{G}(x, \tau) = \begin{cases} 1, & \text{if Sigmoid}(x) > \tau \\ 0, & \text{otherwise} \end{cases} . \tag{6}$$

The values $k_i^{(t)}$ and $v_i^{(t)}$ for token $x^{(t)}$ and attention-head $\text{H}_i$ are retained if $\text{AG}(X)_i^{(t)} = 1$; otherwise, they are discarded. Accordingly, the mask $M_i = \left[m_i^{(s,t)}\right]$ for the attention scores $A_i$ of head $\text{H}_i$ in Equation (4) is defined as:

$$m_i^{(s,t)} = \begin{cases} 0, & \text{if } s < t \\ 1, & \text{if } s = t \\ \text{AG}(X)_i^{(t)}, & \text{otherwise} \end{cases} . \tag{7}$$

Table 4: Performance comparison of Llama2-7B and various KV-Cache eviction strategies **after continual pre-training**. For baselines, $(W_q, W_k, W_v, W_o)$ are made trainable, while in our method, the AG module is also trainable. The eviction ratio is fixed at 50% for baselines such as StreamingLLM and H2O. Notably, our method achieves accuracy exceeding Llama2-7B-cpt, even with over 50% KV-Cache eviction. Higher values indicate better performance for all metrics.

| | Metric | PIQA | ARC-C | ARC-E | RTE | COPA | BoolQ | HellaSwag | Avg. |
|---|---|---|---|---|---|---|---|---|---|
| Llama2-7B-cpt | Acc. | 72.69 | 32.88 | 50.62 | 50.54 | 57.00 | 64.77 | 42.19 | 52.96 |
| StreamingLLM | Acc. | 72.42 | 31.53 | **49.74** | 50.90 | 54.00 | 61.31 | 37.75 | 51.09 |
| H2O | Acc. | 72.20 | 30.85 | 49.38 | **51.99** | 55.00 | **62.42** | 41.45 | 51.90 |
| Ours | Acc. | **76.33** | **32.20** | 48.32 | 50.18 | **59.00** | 60.46 | **64.23** | **55.82** |
| | %Eviction | 43.12 | 46.54 | 45.15 | 48.60 | **55.37** | **50.16** | **61.10** | **50.01** |

| | Metric | MMLU | | Metric | LongBench | |
|---|---|---|---|---|---|---|
| Llama2-7B-cpt | Acc. | 26.64 | | Score | 23.42 | |
| StreamingLLM | Acc. | 26.66 | | Score | 4.61 | |
| H2O | Acc. | 26.45 | | Score | 4.85 | |
| Ours | Acc. | **28.54** | | Score | **13.71** | |
| | %Eviction | **70.36** | | %Eviction | **68.55** | |

In this way, the AG module selectively determines which tokens are retained or discarded for each attention head, based on the global information captured by MHA$'$ and the gating mechanism.

**Computational cost** of processing a sequence through the AG and MHA layers is divided into two parts: the AG module and the MHA module. For the AG module, which processes the input sequence $X \in \mathbb{R}^{n \times d}$, the total FLOPs are:

$$\text{FLOPs}_{\text{AG}} = 3ndd_k'h' + 4n^2 d_k'h' + nh'd_k'h$$

where $d_k'$ is the head dimension of the MHA', and $h'$ is the number of attention heads used in AG. For the MHA module, after the AG processing, only $(1 - t\%)$ of the KV-Cache tokens are retained for attention calculations. The total FLOPs for MHA are:

$$\text{FLOPs}_{\text{MHA after AG}} = 3ndd_kh + 4(1 - t\%)n^2 d_kh + nhd_kd$$

where $d_k$ is the head dimension of the original MHA and $h$ is the number of heads.
Comparing the total FLOPs with and without AG:
Without AG, the original MHA has the following FLOPs:

$$\text{FLOPs}_{\text{original MHA}} = 3ndd_kh + 4n^2 d_kh + nhd_kd \tag{8}$$

With AG, the total FLOPs include both the AG and MHA FLOPs:

$$\text{FLOPs}_{\text{MHA with AG}} = \left(3ndd_k'h' + 4n^2 d_k'h' + nh'd_k'h\right) + \tag{9}$$
$$\left(3ndd_kh + 4(1 - t\%)n^2 d_kh + nhd_kd\right)$$

The key advantage of using AG is that it reduces the number of tokens involved in the attention computation by discarding $t\%$ of the KV-Cache. if $t\%$ is large, the FLOPs reduction from AG is significant, especially when $h' \ll h$, since fewer tokens are processed and the smaller $h'$ may not add substantial overhead.

# B  ADDITIONAL EXPERIMENTS

## B.1  ADDITIONAL RESULTS FOR CONTINUAL PRE-TRAINING

In this section, we perform continual pre-training on Llama2-7B using the same training data and hyperparameter settings as described in Section 4.1.1. For the baselines, we make $(W_q, W_k, W_v, W_o)$ trainable. For our method, the AG module is also made trainable. We have added two datasets, MMLU (Hendrycks et al., 2021) and LongBench (Bai et al., 2023), to further validate the reliability of our method. The results are presented in Table 4.

**Inference efficiency.** In terms of space, the KV-Cache eviction ratio directly reflects the reduction in KV-Cache memory usage. For instance, evicting 50% of tokens reduces KV-Cache storage requirements by half. Our primary focus is on reducing memory usage in this regard.

From a time perspective, the introduction of the AG module does not increase inference latency and can even improve overall efficiency. Consider Llama2-7B, where each Multi-Head Attention (MHA) layer contains 32 heads. By applying the AG module to 4 heads with a 50% eviction ratio, the attention computation in the MHA module is reduced by half. Although the AG module introduces additional computations for 4 heads, this overhead is equivalent to only 1/8 of the original 32-head MHA's computation. Consequently, the total computation with AG is lower than that without AG, as shown in Equation (8) and Equation (9) ($h' = 1/8h, t\% = 50\%, d'_k = d_k$).

## B.2  Results of Continual Pre-training on Mistral

We conducted continual pre-training on Mistral-7B (Jiang et al., 2023) using 5,000 samples from RedPajama (Computer, 2023), and the results are shown in Table 5. Compared to the performance of Llama2-7B presented in Table 1, Mistral's performance slightly declined. We hypothesize that this may be due to the distribution of RedPajama's data being less suited to Mistral. Additionally, this raises the question of whether KV-Cache eviction is model-dependent, and whether its effectiveness is related to the model's expressive power. Although the parameter counts of Mistral-7B and Llama2-7B are similar, Mistral-7B significantly outperforms Llama2-7B. This could suggest that Mistral is utilizing more tokens or scoring them with finer granularity, which results in fewer redundant tokens and thus makes eviction less effective. Furthermore, it is possible that Mistral's use of grouped-query attention (GQA), which inherently involves compression, may make it more challenging to increase the eviction ratio effectively in this context.

Table 5: Performance comparison between Mistral-7B and Ours across various tasks.

|  | Metric | PIQA | ARC-C | ARC-E | RTE | COPA | BoolQ | HellaSwag | Avg. |
|---|---|---|---|---|---|---|---|---|---|
| Mistral-7B | Acc. | 80.09 | 42.37 | 63.14 | 48.01 | 76 | 64.22 | 73.02 | 63.84 |
| Ours | Acc. | 75.90 | 34.24 | 55.2 | 48.01 | 65 | 62.2 | 67.91 | 58.35 |
|  | Eviction | 37.14 | 39.48 | 37.80 | 40.93 | 45.27 | 44.68 | 50.92 | 42.32 |

## B.3  More Visualization

Figure 4 provides a comprehensive view of the layers and attention heads from Figure 3. Additionally, four sample scenarios for the same setup are presented in Figure 5.

## B.4  More Ablation

In this section, we present additional ablation results in Table 6.

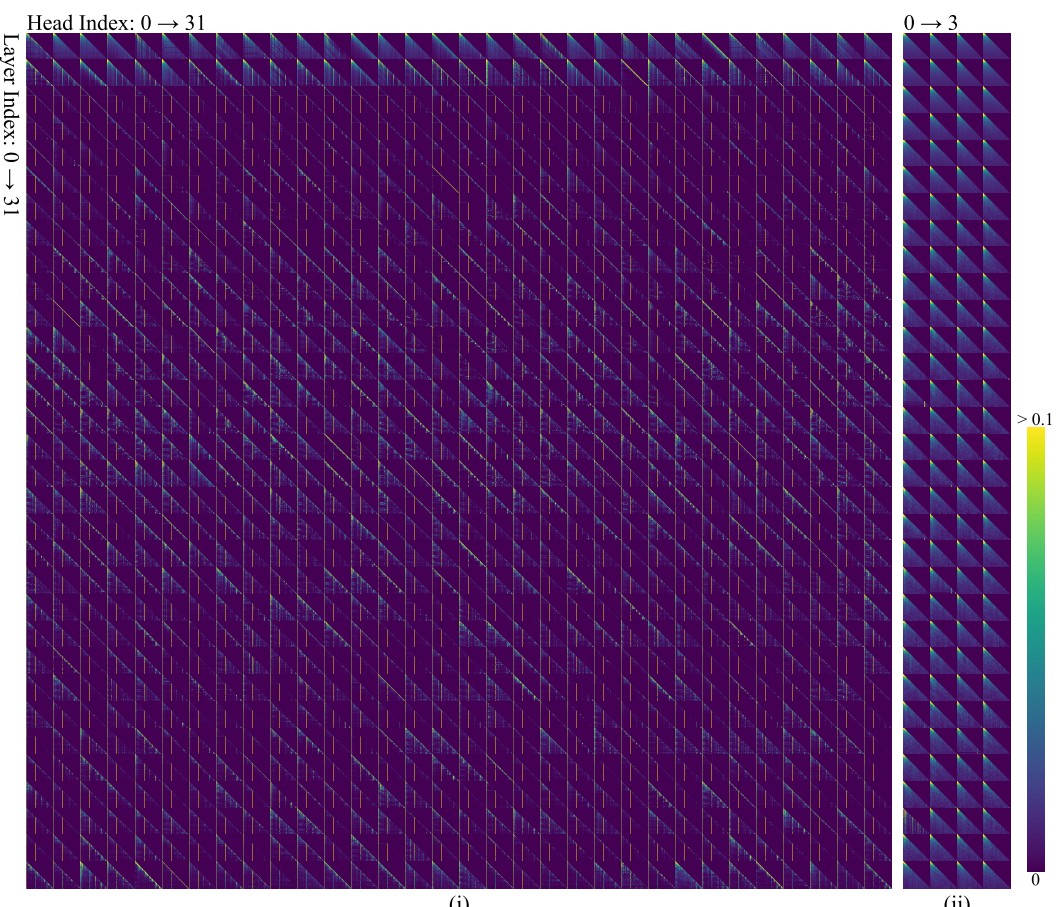

Figure 4: The complete version of Figure 3.

Table 6: Exploring the impact of the number of recent tokens, viewed from the perspective of the attention matrix and considering slanted retention patterns. (1) corresponds to the setup described in Section 4.2, where only the current token is retained, and thus reflecting only the diagonal retention in the attention matrix. For (6-1) and (6-2), the number of recent tokens retained is set to 5 and 10, respectively. The results suggest that increasing the number of recent tokens does not necessarily enhance performance under the AG framework. Further exploration of how to manage recent tokens, such as applying learnable weighted strategies, could be an interesting direction for future work.

|       | Metric    | PIQA  | ARC-C | RTE   | COPA  | BoolQ | OpenBookQA | Avg.  |
|-------|-----------|-------|-------|-------|-------|-------|------------|-------|
| (1)   | Acc.      | 82.15 | 59.66 | 64.26 | 93.00 | 86.82 | 78.80      | 77.45 |
|       | %Eviction | 66.16 | 48.31 | 65.47 | 45.40 | 67.46 | 67.17      | 60.00 |
| (6-1) | Acc.      | 83.08 | 50.85 | 65.34 | 82.00 | 87.31 | 73.20      | 73.63 |
|       | %Eviction | 65.15 | 40.96 | 64.29 | 21.37 | 67.49 | 63.69      | 53.83 |
| (6-2) | Acc.      | 81.61 | 53.56 | 60.29 | 82.00 | 87.37 | 74.20      | 73.17 |
|       | %Eviction | 65.66 | 44.48 | 65.14 | 24.28 | 68.18 | 63.44      | 55.20 |

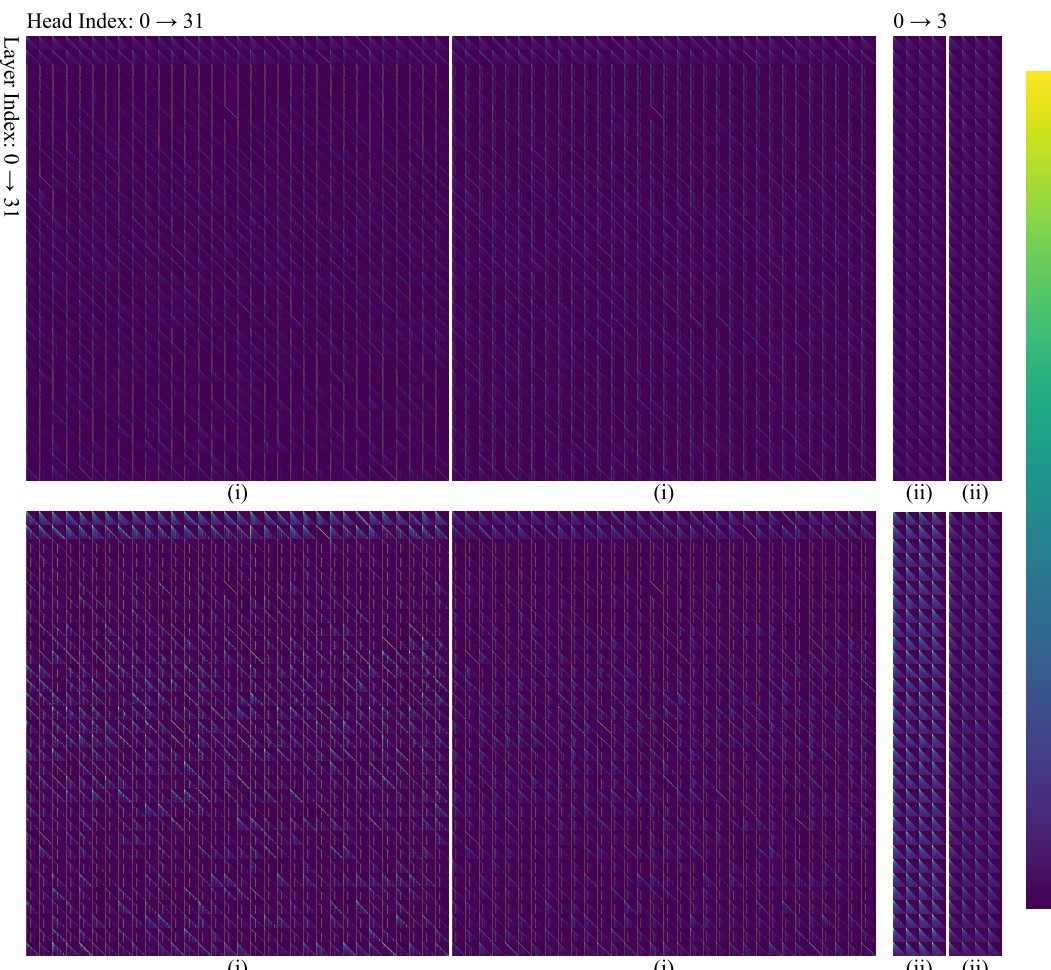

Figure 5: More samples for the same scenario of Figure 3.

