# OpenReview forum: "In-context KV-Cache Eviction for LLMs via Attention-Gate"
_ICLR.cc/2025/Conference — Submitted to ICLR 2025_

### Official Review · Reviewer_bQFG · 2024-10-20

**Soundness:** 3
**Presentation:** 3
**Contribution:** 3
**Rating:** 5
**Confidence:** 4

**Summary:**

This paper introduces a dynamic KV-Cache eviction mechanism called Attention-Gate (AG) for large language models. AG selectively evicts less important tokens based on attention scores, unlike static strategies. Positioned before attention layers, AG can be seamlessly integrated into pre-trained LLMs. The method is validated on various tasks, demonstrating improved efficiency compared to static eviction strategies like StreamingLLM and H2O, with benefits for long-context queries.

**Strengths:**

1. The authors' method is structurally simple yet highly effective, offering more flexibility and better performance compared to previous eviction strategies. It incorporates global information and demonstrates significant improvements in performance.
2. This approach avoids the redundant attention computations seen in prior methods, enabling a more thorough and efficient eviction of tokens.
3. This method requires only a few thousand samples for post-training or fine-tuning to be effective, making it easily applicable to a wide range of existing pre-trained models.
4. The authors provide a detailed analysis of attention behavior across different layers, explaining the effectiveness of their approach and illustrating its specific impact across various cases.

**Weaknesses:**

1. There is no detailed analysis of the data sources used for post-training if other pre-training data besides RedPajama is chosen. There is also no examination of how the amount of post-training data impacts the final results.
2. The method is only effective on Llama-2 and does not show improvement on Mistral-7B, but the paper lacks a detailed analysis of why this is the case. Additionally, there are no experiments on models of other scales besides 7B.
3. The paper lacks analysis on the inference efficiency improvement after eviction, as well as the additional time overhead introduced by the attention-gate computation compared to the original Transformer.
4. It would be helpful to detail the similarities and differences between MoD (https://arxiv.org/abs/2404.02258), which also uses a gate mechanism based on Softmax to select tokens for computation.
5. Most of the evaluation tasks are short-form, few-shot tasks, and there is a lack of comprehensive evaluation tasks that test various LLM capabilities (e.g., MMLU). Additionally, the inclusion of long-form datasets could provide more robust results.
6. There are several hyperparameters involved in AG, but there is no analysis of how these hyperparameters were selected or their impact on performance.
7. Appendices A.2 and A.3 are missing content.

**Questions:**

Please refer to the Weakness.

---

> ### Author Response · Authors · 2024-11-21
> **Response to Reviewer bQFG (Part 1/2)**
>
> Thank you for your detailed and thoughtful comments. Below, we provide our responses to each of your points.
>
> **W1: *Lack of analysis on data sources and impact of post-training data volume.***
>
> **A:**
> Thank you for your feedback. The data used for post-training was randomly sampled from a subset of RedPajama, containing fewer than 5,000 examples. The focus of our experiments is not on the specific content of the data but on demonstrating the efficiency of training our AG module. In our CPT approach, only the AG module is trainable while the original model remains frozen, ensuring that the training data does not affect the base model.
>
> Additionally, we have included a supplementary experiment where all baselines are trained on the same dataset for comparison.
>
> | Model             | Metric       | PIQA     | ARC-C      | ARC-E     | RTE     | COPA     | BoolQ    | HellaSwag     | Avg.      |
> |-------------------|--------------|----------|------------|-----------|---------|----------|----------|---------------|-----------|
> | Llama2-7B-cpt     | Acc.         | 72.69    | 32.88      | 50.62     | 50.54   | 57.00    | 64.77    | 42.19         | 52.96     |
> | StreamingLLM      | Acc.         | 72.42    | 31.53      | **49.74** | 50.90   | 54.00    | 61.31    | 37.75         | 51.09     |
> | H2O               | Acc.         | 72.20    | 30.85      | 49.38     |**51.99**| 55.00    | **62.42**| 41.45         | 51.90     |
> | Ours              | Acc.         | **76.33**| **32.20**  | 48.32     | 50.18   | **59.00**| 60.46    | **64.23**     | **55.82** |
> | Ours              | %Eviction    | 43.12    | 46.54      | 45.15     | 48.60   | **55.37**| **50.16**| **61.10**     | **50.01** |
>
> > For baselines, ($W_q, W_k, W_v, W_o$) are made trainable, while in our method, the AG module is also trainable.
> > The eviction ratio is fixed at 50\% for StreamingLLM and H2O.
> > Higher values indicate better performance for all metrics.
>
> Please refer to Appendix B.1 for more results.
>
> **W2: *Effectiveness limited to llama-2 and lack of analysis on other models and scales.***
>
> **A:**
> Thank you for your comment.
> For Mistral-7B, it cannot evict as aggressively as Llama2-7B, possibly because Mistral uses GQA, which already performs a level of compression that Llama2-7B does not.
> We are currently conducting tests on different model architectures, as well as models of varying scales beyond 7B. The results will be updated as soon as they are available.
>
> **W3: *Missing analysis on inference efficiency and Attention-Gate overhead.***
>
> **A:**
> Thank you for pointing this out.
> Our primary focus is on memory efficiency, as indicated by the KV-Cache eviction ratio (%Eviction). Additionally, we have provided a detailed discussion of how the incorporation of the AG module impacts FLOPs, which can be found in Appendix A (L726–L746) and Appendix B.1 (L756–L765). The actual time overhead is influenced by the kernel design used for implementing AG. We plan to further optimize this design to obtain a more accurate estimate of the time cost.

---

> > ### Author Response · Authors · 2024-11-21
> > **Response to Reviewer bQFG (Part 2/2)**
> >
> > **W4: *Comparison with MoD and its gate mechanism.***
> >
> > **A:**
> > Thank you for pointing out this connection. The key differences between our method and MoD lie in the scope and application of the gating mechanism. In MoD, the gate operates at the token level, scoring each token independently. In contrast, our gate mechanism is sequence-level, utilizing information from the entire sequence to determine the token eviction policy.
> >
> > Additionally, MoD focuses on deciding whether a token should pass through a specific model layer (and in the case of the more advanced MoDE, whether it should pass through a specific FFN within a layer). Our approach, however, is directly applied to the MHA mechanism, addressing the eviction of tokens at this stage.
> >
> > Therefore, our approach is not in conflict with MoD; in fact, it can be combined with MoD for enhanced performance.
> >
> > **W5: *Lack of comprehensive evaluation tasks and long-form datasets.***
> >
> > **A:**
> > Thank you for your suggestion. We have included a supplementary experiment to address this concern.
> >
> > | Model             | Metric       | MMLU     | Metric    | LongBench   |
> > |-------------------|--------------|----------|-----------|-------------|
> > | Llama2-7B-cpt     | Acc.         | 26.64    | Score     | 23.42       |
> > | StreamingLLM      | Acc.         | 26.66    | Score     | 4.61        |
> > | H2O               | Acc.         | 26.45    | Score     | 4.85        |
> > | Ours              | Acc.         | **28.54**| Score     | **13.71**   |
> > | Ours              | %Eviction    | **70.36**| %Eviction | **68.55**   |
> >
> > > For baselines, ($W_q, W_k, W_v, W_o$) are made trainable, while in our method, the AG module is also trainable.
> > > The eviction ratio is fixed at 50\% for StreamingLLM and H2O.
> > > Higher values indicate better performance for all metrics.
> >
> > Please refer to Appendix B.1 for more results.
> >
> > **W6: *Lack of hyperparameter analysis for AG.***
> >
> > **A:**
> > Thank you for this valuable suggestion. We are conducting additional experiments to analyze the impact of hyperparameters on performance. The results will be updated as soon as they are available.
> >
> > **W7: *Missing content in appendices A.2 and A.3.***
> >
> > **A:**
> > Thank you for bringing this to our attention. We have completed and updated the missing content.

---

> > > ### Author Response · Authors · 2024-11-25
> > >
> > > We would like to kindly check if our responses have sufficiently addressed the concerns you raised. If there are any additional points or clarifications needed, we would be happy to discuss them further. Your thoughtful review and feedback are invaluable, and we sincerely thank you for the time and effort you’ve dedicated to improving our work.

---

> > > > ### Comment · Reviewer_bQFG · 2024-11-25
> > > >
> > > > Thank you for your detailed response to my comments. Your explanations have addressed my previous concerns. I will keep my overall rating but will increase the Soundness score from 2 to 3.

---

> > > > > ### Author Response · Authors · 2024-11-26
> > > > > **Reply**
> > > > >
> > > > > Thank you for acknowledging our rebuttal. Nevertheless, we would like to highlight that in our rebuttal and discussions on the OpenReview platform, the focus should be on reaching clear conclusions and providing specific suggestions regarding the submission, leading to a definitive acceptance or rejection. It seems contradictory to suggest that our paper falls below the acceptance threshold while acknowledging its soundness, presentation, and contribution as commendable. We hope the reviewer can reconsider the evaluation if there are no remaining concerns and we welcome any additional feedback.
> > > > >
> > > > > Best regards.

---

### Official Review · Reviewer_rdvv · 2024-10-28

**Soundness:** 2
**Presentation:** 3
**Contribution:** 3
**Rating:** 5
**Confidence:** 4

**Summary:**

This paper is concerned about KV cache compression, especially KV cache eviction. This paper claims that previous parameter-free eviction strategies either can not adapt to the context or can only adapt to the biased context. To remedy this, this paper proposes a parameterized eviction strategy, which learns to evict KV cache at prefilling stage with a trainable module termed Attention-Gate. Experimental results on both continued pretraining and supervised finetuning demonstrate that, with similar eviction proportions, Attention-Gate can achieve better performance in comparison to existing eviction baselines such as StreamingLLM and H2O. Particularly, Attention-Gate combined with LoRA can even realize better performance than LoRA itself. Further ablations indicate that the design choices of Attention-Gate is adequate.

**Strengths:**

- As far as I know, this paper is the first one that introduces parameterized module to achieve KV cache eviction. Previous studies usually rely on parameter-free heuristics.
- This paper is probably one of the first that achieves dynamic KV cache eviction at both layer-level and head-level. Previous studies belong to either layer-level dynamic KV cache eviction (e.g., PyramidKV) or head-level one (e.g., AdaKV).
- The design choices of Attention-Gate are thoroughly elaborated and sufficiently motivated.
- Experimental results show that Attention-Gate outperforms typical baselines with a similar KV cache budget.

**Weaknesses:**

- This paper claims previous studies that can adapt to the given context (e.g., H2O) are largely biased and uses this as a core motivation of a parameterized KV cache eviction strategy (i.e., Attention-Gate). However, there is no clear evidence showcasing that Attention-Gate is not biased. Particularly, from Figure 1, we can see that there are still attention sinks in quite a few heads.
- The proposed strategy in this paper is limited to prefilling stage. There are indeed studies who are also limited to prefilling stage. However, it would be better to consider decoding stage especially in cases where the input prompt is comparably short and the output is long.
- The use of straight-through estimator is not very clear. I recommend more details on this.
- Though it is very nice to see that Attention-Gate combined with LoRA outperforms LoRA, it is not very clear why this would happen. And it is actually very interesting to uncover the reason why eviction can even lead to better results. Perhaps eviction can also serve as a sort of regularization?
- Frustratingly, while KV cache eviction is targeted at improving memory efficiency and potentially latency efficiency, this paper lacks an in-depth comparison on efficiency. And this would predominately affect the contribution and impact of this paper.

**Questions:**

N/A

---

> ### Author Response · Authors · 2024-11-21
> **Response to Reviewer rdvv**
>
> Thanks for your detailed and insightful comments. Below, we provide our responses to your feedback.
>
> **W1: *Bias in attention-gate and presence of attention sinks.***
>
> **A:**
> Thank you for raising this point.
> The core motivation of our method is to develop in-context eviction policies, with addressing attention bias being one of its benefits. The attention bias in prior methods, such as H2O, arises from their reliance on local information to decide which tokens to retain. In contrast, our approach leverages global information by processing the entire sequence through the AG module to generate the eviction policy, effectively avoiding attention bias.
> The attention sinks observed in Figure 1 are a common phenomenon first identified by StreamingLLM. Our method utilizes this phenomenon to improve the eviction process.
>
> **W2: *Limitation to prefilling stage; Consideration of decoding stage.***
>
> **A:**
> Thank you for pointing this out. In fact, our method can also be applied during the decoding phase by storing only the KV-Cache of the AG module. Since the AG module has a limited number of heads, the additional storage requirements would be minimal. We plan to explore this approach further.
>
> **W3: *Clarification on the use of STE.***
>
> **A:**
> Thank you for your question. As mentioned in Section 3.4 (L311–L315), the straight-through estimator (STE) is used to address the non-differentiability issue during training. In the AG module, after applying the sigmoid function, a threshold is used to determine whether to evict tokens, which introduces non-differentiability.
>
> The STE applied in our method can be expressed as follows:
>
> $\text{STE}(x, \tau) = (1_{x \ge \tau} - x).\verb|detach()| + x$
>
> Here, $\tau$ is the threshold, and $\verb|detach()|$ ensures that the first term only affects the forward pass while not contributing to the backward propagation. This allows the training process to proceed smoothly while preserving the intended threshold-based behavior.
>
> **W4: *Understanding why eviction improves performance.***
>
> **A:**
> Thank you for this interesting question. As mentioned in Section 4.2.2 (L421–L422), "*effectively evicting redundant tokens allows the model to focus on relevant information and improves performance.*" A well-designed eviction policy can mask redundant tokens, enabling the model to concentrate more effectively on useful information, which may lead to better outcomes.
>
> Additionally, we agree with your observation that eviction might act as a form of regularization by implicitly reducing noise in the input, which could be an area for further investigation.
>
> **W5: *Lack of in-depth comparison on efficiency.***
>
> **A:**
> Thank you for highlighting this important aspect. We apologize for the omission.
> Our primary focus is on memory efficiency, as reflected in the KV-Cache eviction ratio (%Eviction). We have also included a detailed discussion of how incorporating the AG module affects FLOPs, available in Appendix A (L726–L746) and Appendix B.1 (L756–L765). The actual time overhead depends on the kernel design for implementing AG. We plan to further optimize the kernel design to obtain a more accurate estimate of the time cost.

---

> > ### Comment · Reviewer_rdvv · 2024-11-22
> > **Response to Authors**
> >
> > Your responses have largely addressed my concerns.
> >
> > However, a critical concern still persists, where comparison on practical efficiency is lacking. The eviction ratio for memory and the FLOPs for latency are both theoretical measures.
> >
> > In view of this, I would like to keep my current evaluation.

---

> ### Author Response · Authors · 2024-11-23
> **Response to Reviewer rdvv**
>
> Thank you for your thoughtful comment. We appreciate your concern that theoretical metrics like eviction ratio and FLOPs might not fully reflect practical efficiency.
>
> To provide more clarity, we conducted tests on a single 40GB A100 GPU, comparing inference speed and peak memory usage during the prefiling phase between our implementation (with AG) and the original LLaMA2-7B model (without AG) at various prompt lengths. Below are the results for AG with a 50% eviction ratio:
>
> | Prompt Length | Prefiling Time (s) (w/o AG) | Prefiling Time (s) (w/ AG, 50% Eviction) | Peak Memory (GB) (w/o AG) | Peak Memory (GB) (w/ AG, 50% Eviction) |
> |---------------|-----------------------------|-------------------------------------------|--------------------------|---------------------------------------|
> | 8192          | 2.07                        | 1.60                                      | 33.00                   | 19.40                                |
> | 4096          | 0.70                        | 0.59                                      | 18.78                   | 15.76                                |
> | 2048          | 0.25                        | 0.25                                      | 14.69                   | 14.38                                |
> | 1024          | 0.11                        | 0.19                                      | 13.41                   | 13.69                                |
>
> Note that the model without AG uses the regular self-attention implementation whereas our model with AG is currently implemented with a for loop over the attention heads. Despite this, our method is substantially superior over the baseline, especially for long context cases. We will further develop a kernel-level optimization with triton to avoid the for loop for enhanced efficiency recently.
>
> We hope this practical evaluation addresses your concerns and highlights the real-world benefits of our approach. Thank you again for your valuable feedback.

---

> > ### Author Response · Authors · 2024-11-25
> >
> > May we kindly ask if our responses have sufficiently addressed your concerns? If there are any remaining questions or issues, we would be more than happy to actively discuss them with you. Thank you again for your thoughtful evaluation and the time you have dedicated to reviewing our work. Your feedback is highly appreciated.

---

> > > ### Comment · Reviewer_rdvv · 2024-11-26
> > > **Response to Authors**
> > >
> > > Thanks for providing additional practical measures on both latency and memory.
> > >
> > > However, I still have a question: why the prefill time would vary according to the prompt length?
> > >
> > > In my understanding, the KV cache would be evicted during prefill, but the prefill time would not vary since there are no computation savings.

---

> > > > ### Author Response · Authors · 2024-11-26
> > > > **Clarifying the Reduction in Computation During Prefilling with AG**
> > > >
> > > > Thank you for raising this question. We appreciate your insightful observation and would like to clarify that **our method indeed reduces the computation in the Multi-Head Attention (MHA) module during the prefill stage**.
> > > >
> > > > Specifically, during prefilling, the entire sequence passes through the Attention Guidance (AG) module, which generates an eviction policy. This policy identifies tokens that will ***neither** participate as keys or values in subsequent MHA computations **nor** have their KV caches retained*. As a result, the KV-Cache Eviction Ratio (%Eviction) can effectively reflect the reduction in MHA computation. The detailed computation is provided in Appendix A (L725–L746).
> > > >
> > > > While AG introduces some additional computation, its overhead is minimal because the number of heads in AG is significantly smaller than in the MHA module. For longer sequences, when the MHA eviction ratio becomes significant, this reduction in MHA computation leads to an overall decrease in prefill time.
> > > >
> > > > We hope this explanation addresses your concern. If you have further questions or need additional clarification, we are more than happy to discuss this further. Thank you again for your thoughtful feedback and for helping us refine our work.

---

> ### Comment · Reviewer_rdvv · 2024-11-27
> **Response to Authors**
>
> I acknowledge that the evicted keys or values will not be involved in subsequent computations in prefill stage. However, this is achieved with sparse attention mask if I understand correctly. And I am wondering whether this irregular sparse attention pattern be accelerated through flash attention or something like that?

---

> > ### Author Response · Authors · 2024-11-27
> >
> > Thank you for your thoughtful question.
> >
> > You are correct that our initial implementation uses a sparse attention mask, but this approach does not inherently reduce *actual* computation. While methods like Flash Attention could accelerate sparse patterns, using them would require baseline methods to adopt the same optimization for a fair comparison.
> >
> > To genuinely reduce computation, our method *removes the evicted keys and values directly from the attention computation*, rather than just masking them. This ensures a real reduction in computational cost, independent of specific optimizations like Flash Attention.
> >
> > We hope this addresses your concern. Please feel free to share any further questions or suggestions. Thank you again for your valuable feedback!

---

> > > ### Comment · Reviewer_rdvv · 2024-11-28
> > > **Response to Authors**
> > >
> > > It is good to know keys and values will be instantly evicted even before prefill attention. However, this still raises two questions:
> > >
> > > If keys and values are instantly evicted before attention:
> > > 1. across heads, the number of keys (and values) would be different. I am wondering how this would be achieved?
> > > 2. for query at one position, the corresponding key (and value) might be also evicted. This seems to be a contradiction to the claimed design in this paper.

---

> > > > ### Author Response · Authors · 2024-11-28
> > > >
> > > > Thank you for your thoughtful questions. Please find our responses below:
> > > >
> > > > 1. Handling Different Numbers of Keys and Values Across Heads:
> > > > As we mentioned in our previous response ([see here](https://openreview.net/forum?id=tvQNysCP7C&noteId=nCbr3anPxC)), we process each attention head separately using a for-loop. While this does introduce some time inefficiencies due to the sequential processing, our experiments demonstrate that, particularly for longer texts, our approach actually outperforms the original model in terms of overall speed. Furthermore, although kernel optimization could provide additional speed improvements, this is not the focus of our current work. Our primary aim is to reduce space complexity, which remains our key contribution. That said, we are actively exploring further kernel optimizations to enhance performance.
> > > >
> > > > 2. Eviction of Keys and Values for Each Query:
> > > > Regarding the concern about the potential eviction of keys and values corresponding to a given query, we clarify that this does not contradict the design of our method. As stated in the paper (L131-L132 & L277-L278), we ensure that the diagonal elements (i.e., the key-value pairs corresponding to each query) are always retained. This guarantees that every query can still access its corresponding key and value, even if other keys and values are evicted. This design ensures the correctness of the attention mechanism while also enabling efficient eviction of other less relevant key-value pairs.
> > > >
> > > > We hope this clarifies your concerns. Please feel free to reach out if you have any further questions.

---

### Official Review · Reviewer_noRn · 2024-11-01

**Soundness:** 2
**Presentation:** 2
**Contribution:** 2
**Rating:** 3
**Confidence:** 5

**Summary:**

The paper introduces "Attention-Gate", a parameterized mechanism for KV-Cache eviction in large language models (LLMs). Unlike traditional static or local dynamic eviction methods, Attention-Gate uses contextual inputs to decide which tokens to discard adaptively. Experimental results on continual pre-training and supervised fine-tuning scenarios suggest that the proposed method can achieve better performance than traditional approaches.

**Strengths:**

1. Compared to traditional static or local dynamic eviction methods, Attention-Gate offers a flexible and adaptive approach to KV-Cache eviction in LLMs, enhancing cache management by adjusting to contextual needs.
2. Attention-Gate opts for the attention-like structure to collect contextual information, which can vary among attention heads/layers and be seamlessly plugged into pre-trained LLMs.

**Weaknesses:**

1. Limited architectural detail: The paper lacks a thorough explanation of the detailed design of Attention-Gate, particularly regarding its network architecture and computation equations.
2. Insufficient experimental analysis: The experiments would benefit from more comprehensive analysis to better illustrate the efficiency and effectiveness of Attention-Gate, providing a clearer understanding of its practical impact and performance in different scenarios.

**Questions:**

For the method, the following should be addressed.
1. Currently, it lacks an explanation of the attention-like structure of Attention-Gate, including the network design and computation flow.
2. The only computation equation listed in the paper (Equation 1) is unclear and appears incorrect.
* The definition and computation of $\bar{AG}_l$ are not explained.
* The necessity of introducing an additional threshold $\beta$ is unclear; its role seems to overlap with the probability threshold $\tau$.
* The direct subtraction of $\beta$ seems ineffective for preventing over-aggressive token eviction; an activation or truncation function may be missing.

More experimental analysis can be added.
1. Inference efficiency: including Attention-Gate before each MHA layer adds computational overhead. To assess inference efficiency, a comparison of the computation burden (in FLOPs) between the proposed method and traditional approaches is essential.
2. Comparative evaluation: as the paper mentions other learnable eviction methods like NACL and A2SF, their performance should be included for a more comprehensive comparison of effectiveness.
3. Eviction ratio variability: only a 50% eviction ratio is reported; assessing model performance across varying eviction ratios would provide a more comprehensive analysis.
4. Long sequence analysis: given that KV-cache eviction is critical for long sequences, it would be beneficial to include evaluation tasks involving longer sequences.

---

> ### Author Response · Authors · 2024-11-21
> **Response to Reviewer noRn**
>
> Thank you for your detailed comments. Below, we provide our responses to each of your points.
>
> **W1-Q1: *Lack of explanation for attention-like structure in Attention-Gate.***
>
> **A:**
> Thank you for your suggestion.
> The attention-like structure is a modified version of the standard multi-head attention (MHA) but uses fewer heads. We have added a more detailed formulaic description of the Attention-Gate’s network design and computation flow in Appendix A. For a more in-depth explanation of this attention-like structure, please refer to L680–L687.
>
> **W1-Q2: *Unclear and potential errors in Equation (1).***
>
> **A:**
> We apologize for the typographical errors in Equation (1). These have been corrected in the revised version. We assure you that the implementation in the code is accurate and aligns with the intended design.
>
> **W2-Q1: *Assessment of inference efficiency.***
>
> **A:**
> Thank you for your feedback. Our primary focus is on memory efficiency, as reflected in the KV-Cache eviction ratio (%Eviction). We have also included a detailed discussion of how incorporating the AG module affects FLOPs, available in Appendix A (L726–L746) and Appendix B.1 (L756–L765). Notably, adding AG does not increase the model’s FLOPs.
>
> **W2-Q2: *Inclusion of comparisons with other eviction methods.***
>
> **A:**
> Thank you for your suggestion. Reproducing the baselines presents challenges due to architectural differences; however, we are actively working on it and will update the results as soon as they are available. As highlighted in [Fig. 9](https://www.answer.ai/posts/images/cold_compress/heavy_hitter_consistent.png) of [Cold Compress’s Blog](https://www.answer.ai/posts/2024-08-01-cold-compress.html), H2O outperforms other training-free methods in certain scenarios, reinforcing the adequacy of our current baselines.
> Moreover, the focus of our trainable KV-Cache eviction method differs from that of training-free approaches. Our method focuses on the training phase and developing eviction policies that can automatically adapt to individual downstream tasks based on the context. In contrast, training-free methods are designed to provide an on-average eviction strategies.
>
> **W2-Q3: *Analysis across varying eviction ratios.***
>
> **A:**
> Thank you for your insightful suggestion. Our current method controls the KV-Cache eviction ratio by adjusting hyperparameters, making it challenging to measure performance at a specific eviction ratio. However, as shown in Table 2 of the paper, reducing the eviction ratio generally leads to improved model performance. As a next step, we plan to integrate a controllable component into the Attention-Gate mechanism to enable multiple compression ratios from a single training process.
>
> **W2-Q4 *Evaluation on long sequences.***
>
> **A:**
> Thank you for your suggestion. We have added evaluation results on the Longbench dataset to analyze the performance of our method on long sequences.
>
> | Model             | Metric    | LongBench   |
> |-------------------|-----------|-------------|
> | Llama2-7B-cpt     | Score     | 23.42       |
> | StreamingLLM      | Score     | 4.61        |
> | H2O               | Score     | 4.85        |
> | Ours              | Score     | **13.71**   |
> | Ours              | %Eviction | **68.55**   |
>
> > For baselines, ($W_q, W_k, W_v, W_o$) are made trainable, while in our method, the AG module is also trainable.
> > The eviction ratio is fixed at 50\% for StreamingLLM and H2O.
> > Higher values indicate better performance for all metrics.
>
> Please refer to Appendix B.1 for more results.

---

> > ### Author Response · Authors · 2024-11-25
> >
> > We hope that our responses have sufficiently addressed the points and suggestions raised in your review. If there are any remaining questions or aspects you feel require further clarification, we would be more than happy to discuss them in detail. Thank you once again for your valuable feedback and the effort you have invested in reviewing our work.

---

### Official Review · Reviewer_qk8j · 2024-11-02

**Soundness:** 3
**Presentation:** 3
**Contribution:** 2
**Rating:** 5
**Confidence:** 4

**Summary:**

This paper introduces a simple parameterized KV-Cache eviction mechanism called Attention-Gate, designed to enhance the efficiency of LLMs inference by selectively discarding KV-Cache states for less important tokens. This approach addresses the limitations of traditional static eviction strategies, like those in StreamingLLM, and the attention bias issues in dynamic methods such as H2O. Attention-Gate operates by generating eviction flags for tokens based on the entire context. Experiments showed the performance could be retained with even high eviction ratio compared to baselines.

**Strengths:**

- Attention-Gate is technically sound and easy to implement. The experiment showed its performance in both continued pre-training and lightweight fine-tuning settings with clear visualization of Attention Maps.
- Ablation studies are thoroughly conducted on the impact of the number/dimension of AG heads and the structure choice of AG.

**Weaknesses:**

- The baselines like Local, StreamingLLM, and H2O are weak. As the motivation of Attention-Gate mechanism is to address the attention bias issue, some follow-up works like NACL and A2SF with the same motivation should be included as the baselines to fully show the effectiveness of AG.
- The additional training phase in Attention-Gate may influence the performance of the original LLama2-7B model, so the performance of LLama2-7B with continued training without AG mechanism should be reported.

**Questions:**

- Experiments are conducted on datasets with short input. In practice, the memory issue caused by the KV-Cache is severe in long-text evaluation settings. I would like to see the results reported on Longbench or Infinitebench datasets. This will make the AG more practical and widely adopted.
- I'm confused about the definition of Equation (1): The $\beta$ is task/layer-wise or not? As the authors illustrated in Figure (1), the eviction patterns changed across tasks and layers.

---

> ### Author Response · Authors · 2024-11-21
> **Response to Reviewer qk8j**
>
> Thank you for your detailed comments. Below are our responses to your feedback.
>
> **W1: *Additional relevant baselines.***
>
> **A:**
> Thank you for your suggestion.
> The core motivation of our method is to develop in-context eviction policies, with addressing attention bias being one of its benefits.
> Reproducing the baselines is challenging due to architectural differences, but we are actively working on it and will update results soon. As shown in [Fig. 9](https://www.answer.ai/posts/images/cold_compress/heavy_hitter_consistent.png) of [Cold Compress’s Blog](https://www.answer.ai/posts/2024-08-01-cold-compress.html), H2O outperforms other training-free methods in some scenarios, supporting the sufficiency of our current baselines.
> Moreover, the focus of our trainable KV-Cache eviction method differs from that of training-free approaches. Our method focuses on the training phase and developing eviction policies that can automatically adapt to individual downstream tasks based on the context. In contrast, training-free methods are designed to provide an on-average eviction strategies.
>
> **W2: *Impact of additional training phase on llama2-7b performance.***
>
> **A:**
> Thank you for your suggestion. We have added results where all baselines, including LLaMA2-7B without the AG mechanism, are continual pre-trained on the same dataset.
>
> | Model             | Metric       | PIQA     | ARC-C      | ARC-E     | RTE     | COPA     | BoolQ    | HellaSwag     | Avg.      |
> |-------------------|--------------|----------|------------|-----------|---------|----------|----------|---------------|-----------|
> | Llama2-7B-cpt     | Acc.         | 72.69    | 32.88      | 50.62     | 50.54   | 57.00    | 64.77    | 42.19         | 52.96     |
> | StreamingLLM      | Acc.         | 72.42    | 31.53      | **49.74** | 50.90   | 54.00    | 61.31    | 37.75         | 51.09     |
> | H2O               | Acc.         | 72.20    | 30.85      | 49.38     |**51.99**| 55.00    | **62.42**| 41.45         | 51.90     |
> | Ours              | Acc.         | **76.33**| **32.20**  | 48.32     | 50.18   | **59.00**| 60.46    | **64.23**     | **55.82** |
> | Ours              | %Eviction    | 43.12    | 46.54      | 45.15     | 48.60   | **55.37**| **50.16**| **61.10**     | **50.01** |
>
> > For baselines, ($W_q, W_k, W_v, W_o$) are made trainable, while in our method, the AG module is also trainable.
> > The eviction ratio is fixed at 50\% for StreamingLLM and H2O.
> > Higher values indicate better performance for all metrics.
>
> Please refer to Appendix B.1 for more results.
>
> However, we believe our original comparisons remain fair, as *our CPT approach only trains the AG module while keeping the original model parameters frozen*.
>
> **Q1: *Evaluation on long-text datasets for practicality.***
>
> **A:**
> Thank you for your suggestion. We have added results on the LongBench dataset to evaluate the performance of AG in long-text scenarios.
>
> | Model             | Metric    | LongBench   |
> |-------------------|-----------|-------------|
> | Llama2-7B-cpt     | Score     | 23.42       |
> | StreamingLLM      | Score     | 4.61        |
> | H2O               | Score     | 4.85        |
> | Ours              | Score     | **13.71**   |
> | Ours              | %Eviction | **68.55**   |
>
> > For baselines, ($W_q, W_k, W_v, W_o$) are made trainable, while in our method, the AG module is also trainable.
> > The eviction ratio is fixed at 50\% for StreamingLLM and H2O.
> > Higher values indicate better performance for all metrics.
>
> Please refer to Appendix B.1 for more results.
>
> **Q2: *Clarification on the definition of $\beta$ in Equation (1).***
>
> **A:**
> Thank you for pointing this out. We apologize for the typo, which has been corrected in the revised version. To clarify, $\beta$ is just a hyperparameter for the auxiliary loss during training and is not required during inference.

---

> > ### Comment · Reviewer_qk8j · 2024-11-24
> > **Why is the AVG. score of LongBench with 50% eviction much higher than AG w.o. 50% eviction?**
> >
> > In Appendix Table 4. (Line#720), the 68.55 (AG with 50% eviction) is too much higher than the 13.71 (AG w.o. eviction). Do I misunderstand the meaning of %eviction? Can the authors list the score of sub-tasks in the Longbench to check which sub-task improves most with AG?

---

> > > ### Author Response · Authors · 2024-11-24
> > > **Clarification on %Eviction and LongBench Results in Table 4**
> > >
> > > We apologize for any confusion caused by the presentation of Table 4.
> > >
> > > To clarify, **the last two rows of the table should be read as a single unit**. The first row shows the performance (e.g., LongBench Score) under the corresponding configuration, and the second row indicates the KV-Cache eviction ratio for that configuration. Specifically, for our method, the %Eviction value of 68.55 means that we evicted 68.55% of the KV-cache, and the corresponding LongBench Score is 13.71.
> > >
> > > In contrast, the baseline methods, such as H2O and StreamingLLM, use a fixed eviction ratio of 50%, which is not explicitly listed in the table for simplicity. The purpose of this comparison is to demonstrate that **our method achieves better performance while evicting more KV-cache than these baselines**.
> > >
> > > We hope this explanation clarifies the results and presentation in the table. Thank you for bringing this to our attention.

---

> > > > ### Comment · Reviewer_qk8j · 2024-11-25
> > > > **Thanks for the authors' clarification and active feedback.**
> > > >
> > > > Thanks to the authors' clarification and additional experiments, some of my questions have been addressed, and I have raised my score to 5.
> > > >
> > > > The main concern is still the weak baseline. Since the AG method requires additional training, it should demonstrate significant improvements over strong training-free baselines to justify its broader application.

---

> > > > > ### Author Response · Authors · 2024-11-25
> > > > > **Sincere Appreciation and Emphasis on the Adaptability of Our Trainable KV-Cache Method**
> > > > >
> > > > > Thank you for your insightful review and for raising the score to 5.
> > > > >
> > > > > As we emphasized in our response to W1, our trainable KV-Cache eviction method adopts a different perspective compared to training-free approaches. Our method is designed to achieve adaptive KV-Cache eviction policies at multiple levels, including tasks, model layers, attention heads, and prompts. This level of adaptability is challenging for training-free methods, which are generally aimed at providing a more general-purpose strategy.
> > > > >
> > > > > We sincerely appreciate your thoughtful and comprehensive feedback, which has greatly contributed to refining our work. Thank you again for your valuable comments!

---

### Official Review · Reviewer_TnJW · 2024-11-03

**Soundness:** 2
**Presentation:** 3
**Contribution:** 1
**Rating:** 3
**Confidence:** 5

**Summary:**

This paper aims to reduce the memory overhead of the KV cache in LLM inference. The high-level goal is to ``evict’ ’certain tokens from the attention computation without hurting the model’s accuracy, since their key and value vectors (KV-cache) no longer need to be stored in GPU memory. Differently from previous works that find the evicted tokens through heuristics or attention pattern analysis, this work learns a binary gating function for each attention head that predicts whether a token should be attended to in a context-aware manner. Light training of these gating functions is needed.

Experiments with some tasks suggest that the proposed method can evict a substantial amount of tokens without hurting the model’s accuracy.

**Strengths:**

- The design choices of the proposed method are clearly motivated.
- Presentation is clear and easy-to-follow

**Weaknesses:**

- Questionable choices of the evaluation benchmarks. The paper motivates the importance of KV-cache reduction by stating that “KV-cache can become a bottleneck when dealing with large models and long-context queries.” However, to the best of my knowledge, none of the benchmarks in this paper is “long-context,” nor are the models “large.” Besides, I am not sure benchmarks such as RTE and COPA are the best candidates for properly evaluating these models’ performance, compared to more “up-to-date” choices such as MMLU or BBH.
- Unconventional experimental settings. In the SFT experiment, the paper finetunes the model for each task, which was a common setting for previous-generation models such as BERT. For models such as Llama-2, it is more established to finetune the models with a mixture of a variety of datasets and evaluate the models’ general-purpose capabilities. Therefore, I am not sure whether the conclusions from this experiment would be useful for future research.
- Flawed comparisons and missing baselines. Most of the claimed gains of the proposed method, which requires training, are over zero-shot baselines. A more fair comparison would be to compare to baselines that use the same amount of training as the proposed method. Besides, these two baselines are missing: https://arxiv.org/abs/2308.16137 and https://arxiv.org/abs/2310.01801
- Unjustified claims about efficiency. The paper claims that the proposed method “demonstrate both efficiency and adaptability.” In my opinion, efficiency should be quantified using metrics such as memory overhead, latency, throughput, etc. The paper does not include any of such evaluations in its experiments.
- Limited results that make it unclear whether the proposed method can generalize. With limited selection of the base LLM and evaluation benchmarks, it is unclear whether the proposed method can be useful for models with a different size, architecture (e.g., group-query attention as in Llama-3 and Mistral), and more challenging tasks.
- Potential errors in the training objective: In Eq 1, the purpose of including $\beta$ in Eq 1 is unclear to me since it does have any impact on the gradients. This might be a typo and the paper might have missed a square of the bracket term.

**Questions:**

None

---

> ### Author Response · Authors · 2024-11-21
> **Response to Reviewer TnJW (Part 1/2)**
>
> Thank you very much for your constructive and positive comments. Here are our responses to your comments:
>
> **W1: *Questionable choices of the evaluation benchmarks.***
>
> **A:**
> We apologize for any confusion caused by our choice of evaluation benchmarks. To address this concern, we have added tests on Longbench and MMLU to provide a more comprehensive evaluation.
>
> | Model             | Metric       | MMLU     | Metric    | LongBench   |
> |-------------------|--------------|----------|-----------|-------------|
> | Llama2-7B-cpt     | Acc.         | 26.64    | Score     | 23.42       |
> | StreamingLLM      | Acc.         | 26.66    | Score     | 4.61        |
> | H2O               | Acc.         | 26.45    | Score     | 4.85        |
> | Ours              | Acc.         | **28.54**| Score     | **13.71**   |
> | Ours              | %Eviction    | **70.36**| %Eviction | **68.55**   |
>
> > For baselines, ($W_q, W_k, W_v, W_o$) are made trainable, while in our method, the AG module is also trainable.
> > The eviction ratio is fixed at 50\% for StreamingLLM and H2O.
> > Higher values indicate better performance for all metrics.
>
> Please refer to Appendix B.1 for more results.
>
> **W2: *Unconventional experimental settings.***
>
> **A:**
> Thank you for your valuable feedback. Training on a general dataset followed by evaluation on various datasets closely resembles the scenario of continual pre-training, which we have already explored. For SFT, if a specific downstream task dataset is available, it can be used to train a tailored eviction policy optimized for that particular task. This scenario is also meaningful and worth investigation.

---

> > ### Author Response · Authors · 2024-11-21
> > **Response to Reviewer TnJW (Part 2/2)**
> >
> > **W3: *Flawed comparisons and missing baselines.***
> >
> > **A:**
> > Thank you for highlighting this. We have added results where all methods are trained on the same dataset.
> >
> > | Model             | Metric       | PIQA     | ARC-C      | ARC-E     | RTE     | COPA     | BoolQ    | HellaSwag     | Avg.      |
> > |-------------------|--------------|----------|------------|-----------|---------|----------|----------|---------------|-----------|
> > | Llama2-7B-cpt     | Acc.         | 72.69    | 32.88      | 50.62     | 50.54   | 57.00    | 64.77    | 42.19         | 52.96     |
> > | StreamingLLM      | Acc.         | 72.42    | 31.53      | **49.74** | 50.90   | 54.00    | 61.31    | 37.75         | 51.09     |
> > | H2O               | Acc.         | 72.20    | 30.85      | 49.38     |**51.99**| 55.00    | **62.42**| 41.45         | 51.90     |
> > | Ours              | Acc.         | **76.33**| **32.20**  | 48.32     | 50.18   | **59.00**| 60.46    | **64.23**     | **55.82** |
> > | Ours              | %Eviction    | 43.12    | 46.54      | 45.15     | 48.60   | **55.37**| **50.16**| **61.10**     | **50.01** |
> >
> > > For baselines, ($W_q, W_k, W_v, W_o$) are made trainable, while in our method, the AG module is also trainable.
> > > The eviction ratio is fixed at 50\% for StreamingLLM and H2O.
> > > Higher values indicate better performance for all metrics.
> >
> > Please refer to Appendix B.1 for more results.
> >
> > Besides, we are actively working on reproducing other baselines and will update the results as soon as they are available.
> > However, as shown in [Cold Compress’s Blog](https://www.answer.ai/posts/2024-08-01-cold-compress.html) and specifically in [Fig. 9](https://www.answer.ai/posts/images/cold_compress/heavy_hitter_consistent.png), H2O outperforms other training-free methods, including Hybrid (represented by FastGen [1]), in certain scenarios. Therefore, we believe the current baselines provide sufficient support for our conclusions.
> >
> > Moreover, the focus of our trainable KV-Cache eviction method differs from that of training-free approaches. Our method focuses on the training phase and developing eviction policies that can automatically adapt to individual downstream tasks based on the context. In contrast, training-free methods are designed to provide an on-average eviction strategies.
> >
> > **W4: *Unjustified claims about efficiency.***
> >
> > **A:**
> > Thank you for pointing this out.
> > We primarily focus on memory efficiency, as reflected in the KV-Cache eviction ratio (%Eviction). Additionally, we have included a discussion on the changes in FLOPs after incorporating the AG module, detailed in Appendix A (L726–L746) and Appendix B.1 (L756–L765). Regarding the actual time overhead, it largely depends on the kernel design for implementing AG. We plan to further optimize the kernel design to obtain a more accurate estimate of the time cost, but this is not the primary focus of our current work.
> >
> > **W5: *Limited results that make it unclear whether the proposed method can generalize.***
> >
> > **A:**
> > Thank you for your feedback. We have included additional supplementary experiments, which evaluate performance on MMLU and LongBench. Furthermore, our original evaluation already includes tests on Mistral-7B, as detailed in Appendix B.2. These tests highlight that the use of GQA, which inherently involves compression, may make eviction more challenging compared to Llama2-7B. We hope these experiments further strengthen the validation of our method’s generalization.
> >
> > **W6: *Potential errors in the training objective.***
> >
> > **A:**
> > Thank you for catching this error. You are correct that this was a typo in Eq. 1, which we have corrected in the revised version. We assure you that the experimental setup was implemented correctly, as reflected in the code submitted alongside the manuscript.

---

> > > ### Author Response · Authors · 2024-11-25
> > >
> > > We hope our responses have provided sufficient clarity regarding the concerns raised in your detailed review. If there are any remaining questions or points that require further discussion, we would be glad to address them actively. Thank you again for your thoughtful feedback and the time you’ve invested in evaluating our work—it is greatly appreciated.

---

> > > > ### Comment · Reviewer_TnJW · 2024-11-27
> > > >
> > > > Thanks for the response. The paper has improved with the revisions. However, many of my concerns stand:
> > > > - Re W1: It has been observed that LongBench is not a realistic or challenging benchmark for long-context LMs. I suggest that the authors consider alternatives such as the Needle in a Haystack test (long context retrieval), BABILong (retrieval and reasoning), and RULER (a variety of tasks)
> > > > - Re W2 and W5: I beg to differ with the authors' opinion that SFT can be simulated by their continual pretraining experiments. The latter trains on raw text, while the former aims to elicit the models capabilities in following instructions and solving problems. Despite their same training objectives, they equip the LLMs with distinct capabilities, at least based on the community's current understanding of LLMs. A key challenge in conventional SFT is to trade off different capabilities, which is not reflected in the paper's single-task fine-tuning experiments. Therefore, I do not think the paper has presented sufficient evidence that the proposed technique can generalize to SFT in practice.
> > > > - Re W3: The authors made a great point about the differences between their approach and training-free methods. Have the authors considered comparing to KV-cache reduction method that requires training, which seems a more fair comparison?
> > > > - Re W4: I appreciate the new discussion on FLOPs. However, the authors' promise to further optimize kernel design for improving time efficiency is not convincing to me. See https://arxiv.org/abs/2407.17678

---

> > > > > ### Author Response · Authors · 2024-11-28
> > > > >
> > > > > Thank you for your continued feedback. Here are our responses to your comments:
> > > > >
> > > > > Re W1 (Benchmark choice): While we understand your concerns about LongBench, we believe that it is still a useful benchmark for evaluating token eviction in long-context scenarios. However, we agree that other benchmarks, such as Needle in a Haystack, BABILong, or RULER, would offer valuable insights. We plan to add these in future experiments for a more comprehensive evaluation. Nevertheless, LongBench is still widely regarded as a legitimate test for LLMs' long-context handling.
> > > > >
> > > > > Re W2 and W5 (SFT and generalization): We disagree with your view on the SFT experiments. We are aware that conventional SFT involves a trade-off between tasks, but our method is not meant to replace the complexity of SFT. Our experiments demonstrate how fine-tuning on a specific task with an eviction policy could be beneficial. We have provided evidence that our approach works across multiple benchmarks, and we are continuing to explore how it can generalize to more diverse tasks.
> > > > > As for generalization to SFT, we don’t claim that our method solves every issue in SFT, but rather that it introduces a new perspective on KV-cache eviction that can be adapted to specific tasks.
> > > > >
> > > > > Re W3 (Comparison to KV-cache methods requiring training): As far as we know, there are very few trainable KV-cache eviction methods. A recent approach [1] introduces a learnable mechanism for dropping uninformative tokens, but it faces challenges with batched generation and doesn’t account for continual pre-training or decoding-only LLMs, which are key use cases for our method. We strongly believe that our approach fills this gap and that a fair comparison would involve methods designed for the same context.
> > > > >
> > > > > Re W4 (FLOPs and time efficiency): While we appreciate your skepticism about kernel optimization, we firmly believe that memory efficiency remains the key contribution of our work. Time overhead remains highly dependent on kernel design and system specifics, which we acknowledge requires further exploration. We are actively working on refining this aspect, but the focus of our current work is on memory optimization via KV-cache eviction.
> > > > >
> > > > > We hope these clarifications address your concerns. We remain confident that our approach offers significant advances in memory-efficient LLM inference and believe the revised paper is stronger for it.
> > > > >
> > > > > ---
> > > > > [1] Sotiris Anagnostidis, Dario Pavllo, Luca Biggio, Lorenzo Noci, Aurelien Lucchi, and Thomas Hofmann. Dynamic context pruning for efficient and interpretable autoregressive transformers. Advances in Neural Information Processing Systems, 36, 2024.

---

### Author Response · Authors · 2024-11-21
**General Response**

We thank the reviewers for their thoughtful and constructive feedback. We are encouraged that the reviewers recognized the clear motivation behind our design choices (@TnJW, @rdvv), the technical soundness and flexibility of our approach (@qk8j, @bQFG), and the thoroughness of our ablation studies (@qk8j). Additionally, we appreciate the positive acknowledgment of the novelty of introducing a parameterized approach to KV-Cache eviction (@rdvv).

**We have revised our manuscript and highlighted the modifications in $\color{red}\textbf{red}$.** Below are the major updates made in response to the reviewers’ comments:

1. **Enhanced Description of Attention-Gate**:
   - [@noRn, @bQFG]: We have added a more detailed formulaic description of the Attention-Gate’s network design and computation flow in Appendix A.

2. **Addressing Efficiency**:
   - [@qk8j, @rdvv, @bQFG]: We have included a detailed discussion of inference efficiency, focusing on the FLops with and without AG, in Appendix A (L725-L746).

3. **Correction of Equation (1)**:
   - [@TnJW, @qk8j, @noRn]: We have corrected the typo in Equation (1). Please note that this was a typographical error in the manuscript, and the implementation in our experiments remains unaffected.

4. **Expanded Experimental Results**:
   - [@TnJW, @qk8j, @noRn, @bQFG]: We have added supplementary experiments that include:
     - Results of all baselines after continual pre-training.
     - Evaluation on more challenging benchmarks, including LongBench and MMLU, to demonstrate the generalization of our method.

**Updated Results**:
The following table summarizes the performance of Llama2-7B and various KV-Cache eviction strategies after continual pre-training.
The eviction ratio is fixed at 50\% for StreamingLLM and H2O. Higher values indicate better performance for all metrics.
Please refer to Appendix B.1 for more results.

| Model             | Metric       | PIQA     | ARC-C      | ARC-E     | RTE     | COPA     | BoolQ    | HellaSwag     | Avg.      |
|-------------------|--------------|----------|------------|-----------|---------|----------|----------|---------------|-----------|
| Llama2-7B-cpt     | Acc.         | 72.69    | 32.88      | 50.62     | 50.54   | 57.00    | 64.77    | 42.19         | 52.96     |
| StreamingLLM      | Acc.         | 72.42    | 31.53      | **49.74** | 50.90   | 54.00    | 61.31    | 37.75         | 51.09     |
| H2O               | Acc.         | 72.20    | 30.85      | 49.38     |**51.99**| 55.00    | **62.42**| 41.45         | 51.90     |
| Ours              | Acc.         | **76.33**| **32.20**  | 48.32     | 50.18   | **59.00**| 60.46    | **64.23**     | **55.82** |
| Ours              | %Eviction    | 43.12    | 46.54      | 45.15     | 48.60   | **55.37**| **50.16**| **61.10**     | **50.01** |

| Model             | Metric       | MMLU     | Metric    | LongBench   |
|-------------------|--------------|----------|-----------|-------------|
| Llama2-7B-cpt     | Acc.         | 26.64    | Score     | 23.42       |
| StreamingLLM      | Acc.         | 26.66    | Score     | 4.61        |
| H2O               | Acc.         | 26.45    | Score     | 4.85        |
| Ours              | Acc.         | **28.54**| Score     | **13.71**   |
| Ours              | %Eviction    | **70.36**| %Eviction | **68.55**   |

We believe these updates address the reviewers’ concerns and provide stronger evidence for the effectiveness, adaptability, and efficiency of our proposed method. Thank you again for your insightful comments, which have significantly improved our work.

---

### Meta-Review · Area_Chair_dyfw · 2024-12-18

**Metareview:**

This paper seeks to address the issue of kv-cache sizes growing fast with the size of models and input context. They do so by learning a model component that tries to determine which tokens in the past kv states need to be stored.

While the approach is simple and the eviction detection operates in a data-dependent way, the extent to which it works is a bit unclear. As highlighted by pretty much all reviewers, the choices of baselines as well as of evaluation benchmarks may be less than ideal, but besides that, I would highlight that the evaluation doesn't test for the main claim of the paper which is to enable reductions in memory footprint during inference. Eviction rates are reported, but the extent to which that actually yields savings in inference cost in space remains unclear. There are also some clarity issues, and the approach is itself not clearly defined as pointed out by some of the reviewers. I would encourage the authors to expand the evaluation section and refactor the intro sections with some more detailed description of the proposal and its justification.

**Additional Comments On Reviewer Discussion:**

Reviewers engaged in discussion and mostly highlighted concerns with clarity, especially so for the method definition, and the choice of baselines and evaluation settings. Authors did provided extra results but the manuscript seem to require additional work before being ready for publication.

---

### Decision · Program_Chairs · 2025-01-22

Reject